# Oxicam-type non-steroidal anti-inflammatory drugs inhibit NPR1-mediated salicylic acid pathway

Nobuaki Ishihama[1], Seung-won Choi[1], Yoshiteru Noutoshi [2], Ivana Saska[1], Shuta Asai [1], Kaori Takizawa[1], Sheng Yang He[3,4], Hiroyuki Osada [5] & Ken Shirasu [1,6✉]

Nonsteroidal anti-inflammatory drugs (NSAIDs), including salicylic acid (SA), target mammalian cyclooxygenases. In plants, SA is a defense hormone that regulates NON-EXPRESSOR OF PATHOGENESIS RELATED GENES 1 (NPR1), the master transcriptional regulator of immunity-related genes. We identify that the oxicam-type NSAIDs tenoxicam (TNX), meloxicam, and piroxicam, but not other types of NSAIDs, exhibit an inhibitory effect on immunity to bacteria and SA-dependent plant immune response. TNX treatment decreases NPR1 levels, independently from the proposed SA receptors NPR3 and NPR4. Instead, TNX induces oxidation of cytosolic redox status, which is also affected by SA and regulates NPR1 homeostasis. A cysteine labeling assay reveals that cysteine residues in NPR1 can be oxidized in vitro, leading to disulfide-bridged oligomerization of NPR1, but not in vivo regardless of SA or TNX treatment. Therefore, this study indicates that oxicam inhibits NPR1-mediated SA signaling without affecting the redox status of NPR1.

[1] Plant Immunity Research Group, RIKEN Center for Sustainable Resource Science, Yokohama 230-0045, Japan. [2] Graduate School of Environmental and Life Science, Okayama University, Okayama 700-8530, Japan. [3] Department of Energy Plant Research Laboratory, Michigan State University, East Lansing, MI 48824, USA. [4] Howard Hughes Medical Institute, Michigan State University, East Lansing, MI 48824, USA. [5] Chemical Biology Research Group, RIKEN Center for Sustainable Resource Science, Wako 351-0198, Japan. [6] Graduate School of Science, The University of Tokyo, Bunkyo 113-0033, Japan. ✉email: ken.shirasu@riken.jp

Plants maintain immune mechanisms to restrict the invasion of pathogens. Pathogens have evolved and secreted a plethora of effector molecules to suppress host immune responses and promote colonization. In turn, plants have evolved cognate resistance proteins to recognize these effectors, which leads to a strong immune response, referred to as effector-triggered immunity[1]. This type of immunity is often associated with the hypersensitive response, a localized cell death, to restrict pathogen growth.

Salicylic acid (SA), originally isolated from the bark of white willow (*Salix alba*), has been used for easing pain and reducing fevers for more than two millennia[2,3]. SA derivatives, such as aspirin, acetylsalicylic acid, and other structural analogs, belong to a class of drugs known as non-steroidal anti-inflammatory drugs (NSAIDs) that are commonly used to treat pain and inflammation. The primary target of NSAIDs are cyclooxygenase (COX) enzymes that convert arachidonic acid into prostaglandins in inflammatory processes in humans[4,5]. In plants, SA functions as a key phytohormone that is required for immunity against biotrophic and hemibiotrophic pathogens[6,7]. Many plants produce and accumulate SA at higher levels upon pathogen infection. Ectopic expression of an SA-degrading enzyme or SA deficiency results in the plant's susceptibility to infection, whereas exogenous application of SA potentiates plant defense responses[8–10]. *Arabidopsis* NON-EXPRESSOR OF PATHOGENESIS RELATED GENES 1 (NPR1) is a transcriptional cofactor that regulates the SA-dependent signaling pathway[11,12], and more than 95% of genes responsive to the SA-analog benzothiadiazole are NPR1-dependent[13]. Recently, NPR1 was also shown to function as an adaptor of the Cullin 3 E3 ubiquitin ligase[14].

Recent studies showed that NPR1 and its paralogs, NPR3 and NPR4, have affinity for SA and are involved in SA perception[15–19]. Because SA promotes the NPR1–NPR3 interaction but disrupts the NPR1–NPR4 interaction, NPR3 and NPR4 are proposed to function as adaptors of the Cullin 3 E3 ubiquitin ligases for the turnover of NPR1 in response to SA[15]. Structural analysis of the NPR4–SA complex showed that SA is completely buried in the hydrophobic core of NPR4, thus causing its drastic conformational change, which presumably disrupts the NPR1–NPR4 interaction[19]. Amino acid residues forming the interaction surface of NPR4 with SA are highly conserved in NPR1 paralogs, corroborating that NPR1 also binds to SA[19]. In contrast, Ding et al.[18] demonstrated that NPR3 and NPR4 are transcriptional co-repressors of SA-responsive genes, but that repression is lost upon SA binding. Further, the same study reported that SA binding to NPR1 promotes its transcriptional activation[18].

*Arabidopsis* NPR1 is a cysteine-rich protein whose activity is regulated by posttranslational modifications at cysteine residues[20]. In particular, Mou et al.[21] reported that, under normal conditions, NPR1 exists predominantly in cytoplasm as an oligomer that is formed via redox-sensitive intermolecular disulfide bonds between cysteine residues. Later, Tada et al.[22] found that an NO donor, *S*-nitrosoglutathione, promotes *S*-nitrosylation of $Cys^{156}$, which in turn facilitates disulfide-bridged NPR1 oligomerization. SA accumulation results in a shift in cellular redox balance toward the reductive state, which then leads to the release of active NPR1 monomers by thioredoxin-mediated reduction of the disulfide bonds[21,22]. Thus, NPR1 has been proposed to be a redox sensor in plants.

Previously, we established a high-throughput screening method by employing an *Arabidopsis* suspension cell-*Pseudomonas syringae* pv. *tomato* DC3000 (*Pto*) *avrRpm1* system to identify plant immune-priming agents[23–28]. In this system, the suspension cells exhibit immunity-associated programmed cell death triggered by recognition of the effector protein AvrRpm1 (ref. [29]). Plant immune-priming compounds, such as SA and tiadinil, potentiate this type of effector-triggered cell death in suspension cells at 10 μM concentration[28]. By screening a set of chemical libraries, we isolated several plant immune-priming compounds that enhance effector-triggered cell death in this assay[23–26].

Here, we report that oxicams, a class of NSAIDs, potentiate effector-triggered cell death but, unlike SA, inhibit immunity against bacteria and down-regulate SA-dependent immune responses in plants. Among them, tenoxicam (TNX) disturbs the SA-induced cellular redox shift, which is important for NPR1 homeostasis, and broadly suppresses SA-responsive genes and reduces NPR1 levels independently of NPR3 and NPR4. Notably, our biochemical evidence revealed that the predominant form of NPR1 in vivo is a reduced one regardless of SA or TNX treatment, not supporting the previously proposed oligomer–monomer transition model of NPR1.

## Results

### NSAIDs potentiate effector-triggered cell death in suspension cells.

In the previous screening assays using *Pto avrRpm1*-induced cell death in *Arabidopsis* suspension cell cultures[26], we identified 19 NSAID compounds as cell death potentiators. Eight NSAID compounds were from the MicroSource library and 13 compounds were from the NPDepo library[30], including ibuprofen (IBF) and ketoprofen (KPF), which were included in both libraries (Supplementary Fig. 1 and Supplementary Table 1). We selected eight representative compounds in terms of chemical structure for further studies. Similar to aspirin, *Pto avrRpm1*-induced cell death was significantly enhanced in a dose-dependent manner by the application of propionic acid derivatives (IBF and naproxen (NPX)), acetic acid derivatives (indomethacin (IDM) and sulindac (SLD)), anthranilic acid derivatives (mefenamic acid (MFA)), and oxicam derivatives (TNX, meloxicam (MLX) and piroxicam (PRX)), although some were toxic at higher concentrations in mock-treated controls (Supplementary Fig. 2).

### Oxicams suppress immunity to *Pto* in *Arabidopsis*.

SA has been shown to enhance plant immunity[6,7,9]; therefore, we next tested whether NSAIDs could also affect disease resistance in *Arabidopsis*. Soil-grown *Arabidopsis* plants were inoculated with the virulent strain *Pto* with 100 μM NSAID compounds and bacterial growth was measured. Surprisingly, immunity against *Pto* was significantly suppressed by application of the oxicams TNX, MLX, and PRX, while the non-oxicams IBF, IDM, and MFA did not show the suppressing effect (Fig. 1a). Oxicam derivatives did not promote bacterial growth in vitro without host plants at the concentration of 100 μM (Supplementary Fig. 3), suggesting that these compounds do not directly promote the growth per se. Importantly, at 100 μM, oxicam derivatives alone did not cause visible cell death in *Arabidopsis* leaf tissues, but potentiated *Pto avrRpm1*-induced cell death (Fig. 1b, c). We conclude that, while both oxicam-type and non-oxicam-type NSAIDs potentiate effector-triggered cell death in suspension cell cultures, only oxicam-type NSAIDs suppressed host immunity in *Arabidopsis* plants.

### TNX suppresses SA-responsive genes.

Given that the oxicam-type NSAIDs and SA share a common target in animal cells, but unlike SA are able to suppress immunity against *Pto*, we focused on TNX, which is the most potent compound among oxicam derivatives tested in the cell death assay (Fig. 2a and Supplementary Fig. 2), to elucidate their mode or modes of action. First, we tested the effect of TNX on representative marker genes, including *PR1* for SA-dependent immune signaling and *PDF1.2*

for jasmonic acid-dependent immune signaling in leaf tissues in response to *Pto* infection. Application of TNX significantly suppressed *Pto*-induced *PR1* expression at 24 h after inoculation (Fig. 2b), but increased *Pto*-induced *PDF1.2* expression (Fig. 2c). We also assessed the effect of TNX on SA-induced *PR1* expression in Col-0 wild-type (WT) seedlings, and observed its

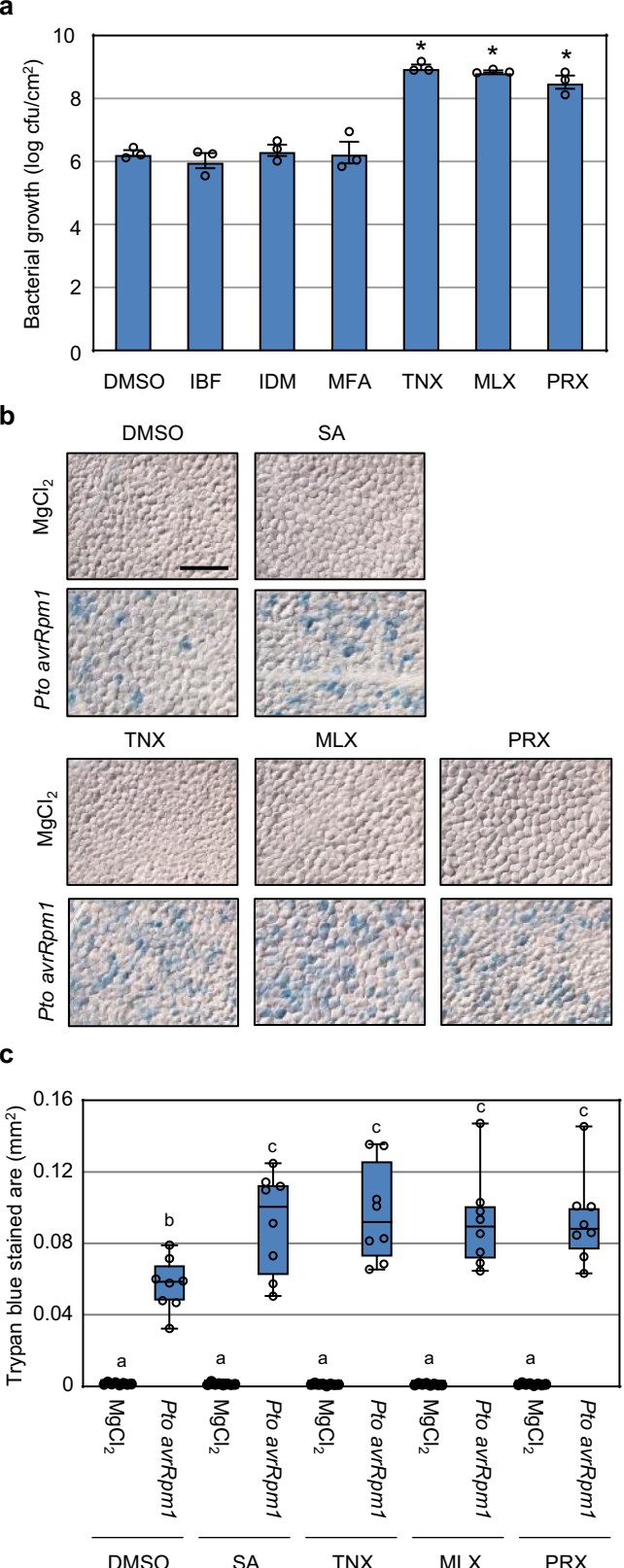

**Fig. 1 Effect of NSAIDs on *Pto* growth in plants. a** Col-0 WT plants were syringe-infiltrated with $1 \times 10^5$ cfu/mL of *Pto* with indicated compounds at 100 μM, or with 0.5% DMSO as negative control. Bacterial growth was determined 4 days post-inoculation. Data are shown as mean ± SE ($n = 3$ biological replicates). Asterisks indicate significant differences from DMSO (two-sided Dunnett test, $p < 0.05$). Experiments were repeated three times with similar results. **b, c** Col-0 WT plants were treated with 100 μM chemicals 3 days before infection. *Pto avrRpm1* suspension ($1 \times 10^7$ cfu/mL in 10 mM $MgCl_2$) was infiltrated into the abaxial side of leaves and the plants were incubated for 6 h. Dead cells were stained with Trypan blue and observed under a microscope (**b**). Bar = 200 μm. The stained area was measured using an imaging software. Two leaves were taken from each of four individual plants. Center lines indicate the medians; box limits indicate the 25th and 75th percentiles. Whiskers represent the minima and maxima. Different letters indicate significant differences (two-sided Tukey–Kramer test, $p < 0.05$).

suppression in a dose-dependent manner (Fig. 2d). Consistently, MLX and PRX treatments of transgenic *Arabidopsis* plants carrying *beta-glucuronidase* (*GUS*) under the regulatory control of the *PR1* promoter[31] suppressed SA-induced *PR1* expression similar to TNX treatment (Fig. 2e). Bacterial growth assays in the *sid2-2* and *npr1-1* mutants, which are defective in SA biosynthesis and signaling, respectively[10,11], revealed that the application of TNX does not further increase susceptibility to *Pto* (Fig. 2f). These data suggest that the oxicam-type NSAIDs interfere with the SA signaling pathway, resulting in reduced immunity against *Pto*.

**TNX represses almost half of SA-induced NPR1-dependent genes.** We compared transcriptome profiles between Col-0 WT and *npr1-1* seedlings treated with either (i) mock, (ii) 100 μM SA, (iii) 100 μM TNX, or (iv) 100 μM SA + 100 μM TNX (Supplementary Data 1). Comparison between TNX-treated Col-0 WT and mock-treated Col-0 WT showed that 2402 genes were induced (FDR = 0.001 and $\log_2$ fold change ($\log_2$FC) > 1) and 2662 were repressed (FDR = 0.001 and $\log_2$FC < −1) by TNX treatment (Fig. 3a and Supplementary Data 2 and 3). Gene ontology (GO) analysis revealed that there was significant overrepresentation of environmental stress-related GO terms among TNX-induced genes, including response to abiotic stimulus (GO:0009628), response to inorganic substance (GO:0010035), two oxidative stress-related GO terms, response to oxygen-containing compound (GO:1901700), and response to oxidative stress (GO:0006979) (Fig. 3b). In addition, there were two overrepresented GO terms related to the biosynthesis of flavonoids, which are plant secondary metabolites with antioxidant activity and are responsive to biotic and abiotic stresses[32] (GO:0009812 and GO:0009813) (Fig. 3b). Response to abiotic stimulus (GO:0009628) was also significantly overrepresented among TNX-repressed genes. These results suggest that TNX broadly stimulates the expression of environmental or oxidative stress-related genes, while negatively affecting the expression of an independent set of stress-response genes (Fig. 3b).

By comparing SA- and mock-treated Col-0 WT (SA/mock in WT), we found 940 genes were induced by SA treatment (FDR = 0.001 and $\log_2$FC (SA/mock in WT) > 1) and classified them as "SA-inducible" (Fig. 3c and Supplementary Data 4). Upon SA treatment, two thirds (67.1%; 631 genes) of "SA-inducible" genes in the *npr1* mutant displayed fold-induction levels that were less than half of the Col-0 WT ($\log_2$FC (SA/mock in WT) – $\log_2$FC (SA/mock in *npr1*) >1) (Fig. 3c and Supplementary Data 5). Our data confirm that NPR1 is a master regulator of SA signaling in plants. Thus, we classified this subset

of genes as "NPR1-dependent". In addition, comparison of SA- and SA + TNX-treated Col-0 WT identified 383 genes which we classified as "TNX-sensitive" since their expression was suppressed upon co-treatment with TNX (log₂FC (SA/SA + TNX) > 1) (Fig. 3c and Supplementary Data 6).

We then categorized the "NPR1-dependent" genes into three groups. Group I contained 313 genes whose expression was significantly decreased by co-treatment with TNX (log₂FC (SA/SA + TNX in WT) > 1), Group II contained 218 genes whose expression was not significantly suppressed by co-treatment with TNX, nor up-regulated by TNX treatment alone (log₂FC (SA/SA + TNX in WT) < 1 and log₂FC (TNX/mock in WT) < 1), and Group III consisted of 100 genes whose expression was not suppressed by co-treatment with TNX, but was up-regulated by TNX treatment alone (log₂FC (SA/SA + TNX in WT) < 1 and log₂FC (TNX/mock in WT) > 1) (Fig. 3c, d, e and Supplementary Data 7). Nearly half (49.6%) of SA-inducible NPR1-dependent genes (Group I) were repressed following co-treatment with TNX (Fig. 3d, e). Consistently, PR1 and WRKY transcription factor genes, WRKY38, WRKY53, and WRKY62, which are directly controlled by NPR1 (refs. [13,33]), were included in Group I (Supplementary Fig. 4). These results show that TNX broadly affects SA-inducible NPR1-dependent genes. However, 207 genes (32.8%) of SA-inducible NPR1-dependent genes were suppressed upon co-treatment with TNX even in the npr1 mutant (log₂FC (SA/SA + TNX) in npr1 > 1), implying that TNX also affects NPR1-independent processes (Supplementary Data 8). Comparison between SA- and mock-treated Col-0 WT also identified 256 genes as SA-repressed genes (FDR = 0.001 and log₂FC < −1) (Supplementary Fig. 5 and Data 9). Among the SA-repressed genes, expression of 142 genes (55.4%) in the npr1 mutant and 43 genes (16.8%) in Col-0 WT co-treated with TNX were restored (Supplementary Fig. 5 and Data 10, 11). Interestingly, 88% (38/43 genes) of genes restored by TNX treatment were overlapped with those in the npr1 mutant.

**TNX suppresses nuclear NPR1 accumulation in response to SA.** Because nearly half of NPR1-dependent SA-inducible genes were repressed by TNX, we examined the effect of TNX treatment on subcellular localization and accumulation of NPR1 using a transgenic Arabidopsis line expressing NPR1-YFP under the control of its native promoter in the npr1-6 knockout mutant background (NPR1p:NPR1-YFP)[34]. As had been reported previously[34], 100 µM SA treatment increased nuclear YFP fluorescence, whereas no fluorescence was detected in either the cytoplasm or nuclei in the absence of SA (Fig. 4a). Under our experimental condition, we did not observe the formation of SA-induced NPR1 condensates previously detected in the NPR1-GFP overexpressing seedlings after 5 mM SA treatment[14]. Interestingly, SA-inducible nuclear YFP fluorescence was consistently lower when oxicam derivatives were added to SA treatments (Fig. 4a). Immunoblot analysis using anti-NPR1 antibody, which were raised against recombinant NPR1 (Supplementary Fig. 6), revealed that application of the oxicam-type NSAIDs consistently suppressed SA-induced NPR1-YFP accumulation (Fig. 4b). Conversely, non-oxicam-type NSIADs IBF and IDM did not suppress SA-induced NPR1-YFP accumulation (Supplementary Fig. 7). NPR1 transcripts were slightly, but not significantly suppressed by TNX in SA-treated plants (Supplementary Fig. 8), so the reduction of NPR1 levels by the oxicam-type NSAIDs is likely due to a post-transcriptional regulatory mechanism.

As mentioned above, a previous study showed that NPR3 and NPR4 are SA receptors and involved in degrading NPR1 as part of a Cullin 3 E3 ubiquitin ligase[15]. To test whether NPR3 and NPR4 are involved in TNX-dependent inhibitory effects on

NPR1, we measured NPR1 protein levels in the npr3npr4 mutant by immunoblot. Endogenous NPR1 levels were increased by SA treatment as had been reported previously[14,33] and were suppressed by co-treatment with TNX, consistent with the observation in NPR1p:NPR1-YFP (Fig. 4b, c). In the npr3npr4 double mutant, both NPR1 and PR1 accumulated 24 h after SA treatment, and TNX suppressed this effect (Fig. 4c), indicating that these phenomena are independent of NPR3 and NPR4. In the absence of SA, NPR1 accumulation was not observed in the npr3npr4 mutant, unlike the previous report[15]. We also tested the effect of TNX treatment on the interactions between NPR1 and NPR3, as well as NPR1 and NPR4, with a yeast two-hybrid assay. Disruption of the NPR1–NPR4 interaction by SA in yeast was confirmed as described previously[15,35] (Supplementary Fig. 9a, b). Conversely, in contrast to Fu et al.[15], SA application did not promote NPR1–NPR3 interaction (Supplementary Fig. 9a). Instead, our data are consistent with other recent reports by Castello et al.[35] and Ding et al.[18] that do not show NPR1–NPR3 interactions in the presence of SA. Importantly, TNX treatment did not result in any apparent effects in any combination of NPR1, NPR3, and NPR4 interactions examined (Supplementary Fig. 9a). These results indicate that neither NPR3 nor NPR4 is involved in the TNX effect on NPR1 protein levels.

**TNX inhibits SA-induced cellular redox shift.** Previous studies indicate that a cellular redox shift is important for both activation and nuclear accumulation of NPR1 (ref. [21]). We therefore investigated whether TNX influences the SA-inducible cellular redox shift. For this purpose, we measured total glutathione (GSH) content and the ratio of reduced (GSH) to the oxidized (GSSG) form in SA or SA + TNX-treated Arabidopsis seedlings. SA treatment increased both the total GSH amount and the GSH/GSSG ratio, both of which were suppressed by TNX co-treatment (Fig. 5a, b). In addition, TNX alone reduced the GSH/GSSG ratio, indicating that TNX treatment alters the cytoplasmic redox status.

To determine whether TNX alters the redox state of the cysteine residues in NPR1, we employed a thiol-specific labeling strategy using a 2 kDa polyethylene glycol-conjugated maleimide (PEG-Mal) that selectively reacts with free thiol groups and adds the PEG polymer per modified cysteine. Addition of PEG-Mal to a cysteine results in an SDS-PAGE gel mobility shift in proportion to the number of PEG moieties added[36]. Before labeling with PEG-Mal, proteins were acidified and denatured with trichloroacetic acid (TCA)/acetone to prevent further oxidation of cysteine thiols in vitro[37,38]. First, we assessed the resolution of the PEG-Mal labeling assay by monitoring NPR1-GFP, which contains 19 cysteines in total (17 and 2 cysteines in NPR1 and GFP, respectively). GFP-tagged Arabidopsis NPR1 (NPR1^WT-GFP) and its cysteine mutant (NPR1^C82AC216A-GFP) were expressed in Nicotiana benthamiana leaves, and total proteins were reduced by a reducing agent tris (2-carboxyethyl) phosphine (TCEP), followed by labeling with PEG-Mal. Non-labeled NPR1^WT-GFP and NPR1^C82AC216A-GFP exhibited a comparable mobility and were observed around the predicted molecular mass (93 kDa) (Fig. 5c). However, we detected the PEGylated NPR1-GFPs around 150 kDa, even though the theoretical molecular mass of NPR1-GFP with 19 PEG polymers is estimated to be 131 kDa. This discordance is likely due to the absence of a charge on PEG, such that PEG-labeling can potentially affect the charge to mass ratio, and thus its mobility in SDS-PAGE[39]. Most importantly, PEGylated NPR1^C82AC216A-GFP migrated further than PEGylated NPR1^WT-GFP (Fig. 5c), demonstrating that the PEG-Mal labeling assay has enough resolution to detect one disulfide-bond formation (or reduction) in 19 cysteine residues.

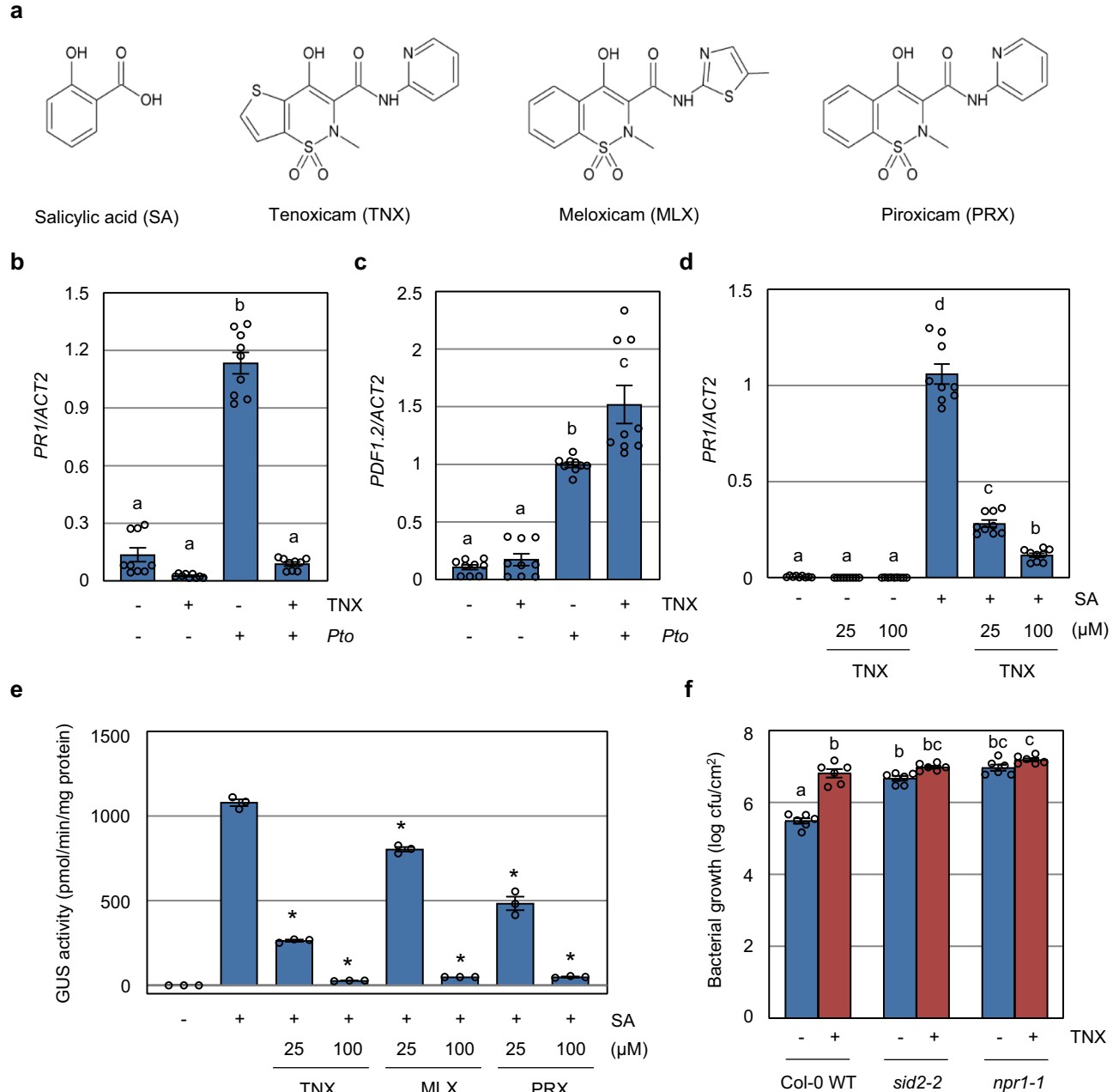

**Fig. 2 Oxicam compounds inhibit SA-dependent immunity. a** Chemical structures of salicylic acid (SA), tenoxicam (TNX), meloxicam (MLX), and piroxicam (PRX). **b**, **c** Col-0 WT plants were syringe-infiltrated with (+) or without (−) $1 \times 10^5$ cfu/mL *Pto* suspension in the presence (+) or absence (−) of 100 μM TNX for 24 h. *PR1* (**b**) and *PDF1.2* (**c**) transcript levels were measured by qRT-PCR with cDNA prepared from each of the samples. The expression values of individual genes were normalized against *ACT2*. Data are shown as means ± SE of three independent experiments, each with three replicates. Different letters indicate significant differences (two-sided Tukey–Kramer test, $p < 0.05$). **d** Col-0 WT seedlings were treated with TNX for 1 h before treatment with 100 μM SA (+) or water (−) for 24 h. *PR1* transcript levels were measured by qRT-PCR. Expression values were normalized against *ACT2*. Data are shown as means ± SE of three independent experiments, each with three replicates. Different letters indicate significant differences (two-sided Tukey–Kramer test, $p < 0.05$). **e** *PR1p:GUS* seedlings were treated with indicated concentration of oxicam derivatives (TNX, MLX, or PRX) or 0.5% DMSO control for 1 h before treatment with 100 μM SA (+) or water (−). GUS activity was measured after 24 h. Data are shown as means ± SE ($n = 3$ biological replicates). Asterisks indicate significant differences from 100 μM SA alone (two-sided Dunnett test, $p < 0.05$). **f** Col-0 WT, *sid2-2*, and *npr1-1* plants were syringe-infiltrated with $1 \times 10^5$ cfu/mL of *Pto* suspension in 10 mM MgCl$_2$ and 0.25% DMSO in the presence (+) or absence (−) of 100 μM TNX. Bacterial growth was determined 3 days-post-inoculation. Data are shown as means ± SE ($n = 6$ biological replicates). Different letters indicate significant differences (two-sided Tukey–Kramer test, $p < 0.05$). In **e** and **f**, experiments were repeated three times with similar results.

To monitor the redox state of the cysteine residues in NPR1 in intact plants, we employed *35Sp:NPR1-GFP* (in the *npr1-1* background) plants which accumulate higher amount of NPR1-GFP regardless of SA treatment[40]. Total proteins of SA and/or TNX-treated *35Sp:NPR1-GFP* plants were incubated with or without

TCEP, and then were labeled with PEG-Mal (Fig. 5d). In addition, we specifically labeled cysteines forming disulfide bond by blocking free cysteine thiols with a thiol-alkylating agent, *N*-ethylmaleimide (NEM) prior to PEG-Mal labeling (Fig. 5d). NEM forms covalent bonds with free thiol groups, but not with pre-formed disulfide bonds

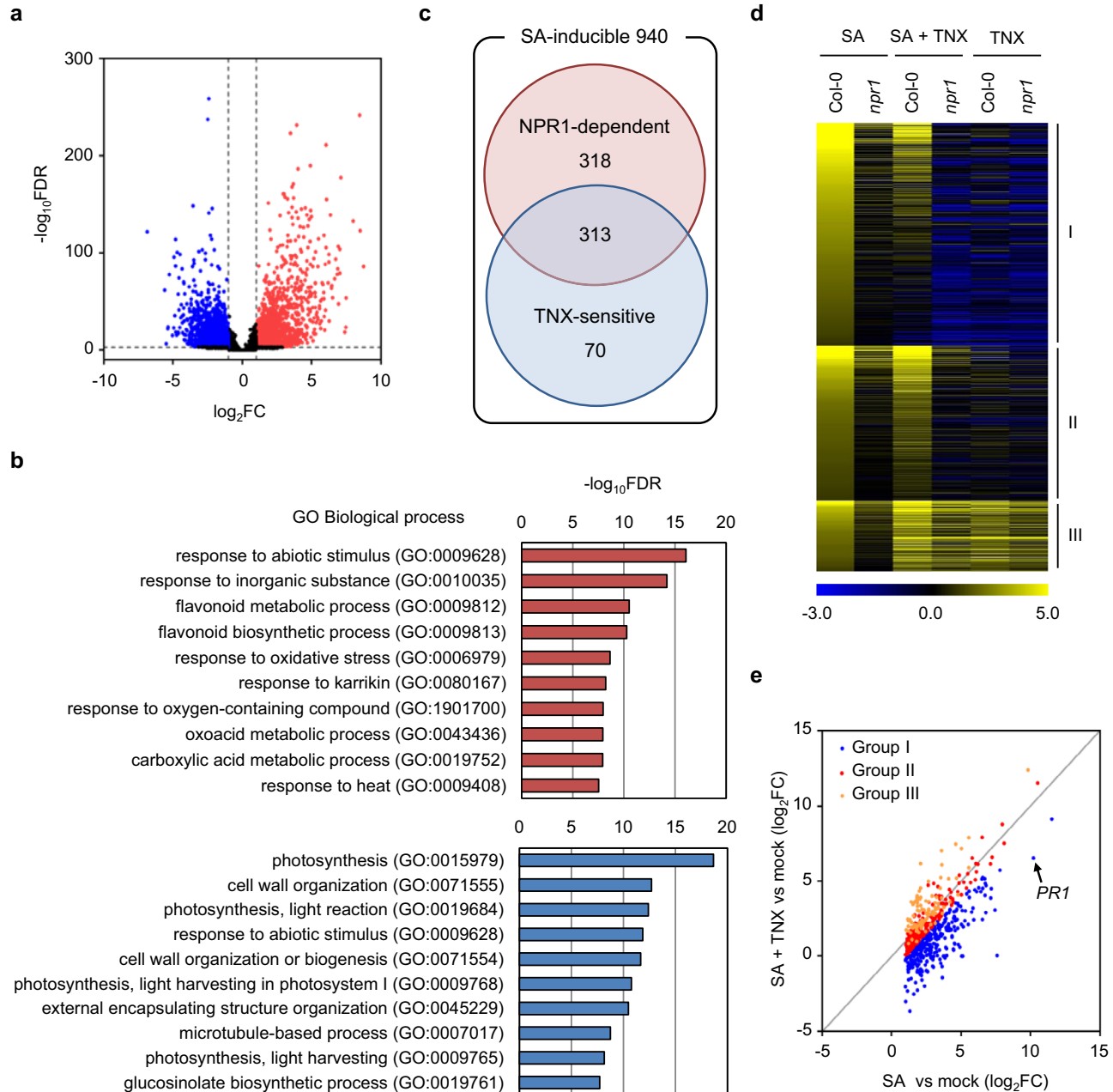

**Fig. 3 TNX broadly affects SA-induced NPR1-dependent genes. a** Volcano plots representing global changes in gene expression by TNX treatment in Col-0 WT. Red and blue dots indicate significantly up-regulated (FDR < 0.001 and $\log_2$ fold change (FC) > 1) and down-regulated genes (FDR < 0.001 and $\log_2$FC < −1), respectively. **b** Gene ontology (GO) enrichment analysis of TNX-induced genes (upper panel) and TNX-repressed genes (lower panel). The top 10 significantly enriched biological process GO terms are shown with their −$\log_{10}$FDR. **c** Venn diagram showing the number of SA-inducible genes whose expression levels were reduced in the *npr1* mutant (NPR1-dependent) and/or co-treatment with TNX (TNX-sensitive). **d** Heat map depicting $\log_2$ (fold change) of SA-inducible and NPR1-dependent genes in Col-0 WT or *npr1* mutant treated with SA or SA + TNX. Genes were categorized into three groups based on expression patterns as described in the "Results" section. Yellow and blue colors indicate up- and down-regulated expression levels, respectively, compared with mock-treated plants. **e** Scatter plot showing expression levels of SA-inducible and NPR1-dependent genes in SA- and SA + TNX-treated Col-0 WT. The *x*- and *y*-axis represent the fold changes between SA-treated and mock-control (SA vs mock) and SA + TNX-treated and mock-control (SA + TNX vs mock), respectively. Blue, red, and orange dots represent group I, II, and III genes, respectively. The arrow-head indicates the spot of *PR1*.

in the protein, thus preventing free thiol groups from further chemical reaction[37,38]. Without reduction, PR1 in PEG-Mal-labeled samples migrated as a 15 kDa polypeptide similar to unlabeled PR1 as expected since PR1 contains six cysteines that form three intramolecular disulfide bonds and no free thiols (Fig. 5e, f)[41]. In contrast, disruption of the disulfide bonds in PR1 with TCEP prior to PEG-Mal labeling resulted in a clear mobility shift, confirming that

PEG-Mal selectively reacts with free thiols. The difference in PR1 signals between samples could be due to the cleavage of the disulfide bond that affects antigenicity to the anti-PR1 antibody. SA or SA + TNX treatment did not affect protein band migration patterns, indicating that the redox state of cysteines in NPR1 are not affected by SA or SA + TNX treatment (Fig. 5e, f). Notably, reduction of NPR1-GFP with TCEP prior to PEG-Mal labeling also

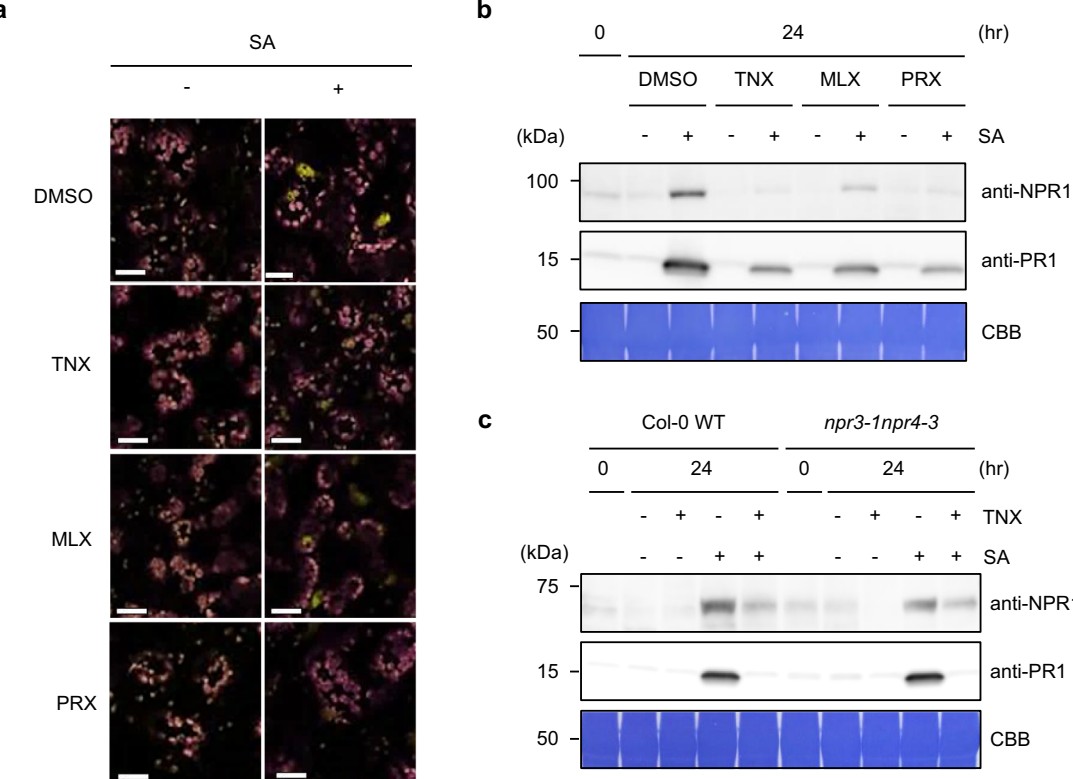

**Fig. 4 TNX inhibits nuclear NPR1 accumulation in response to SA. a** *NPR1p:NPR1-YFP* seedlings were treated with 100 μM oxicam derivatives (TNX, MLX, or PRX) or 0.5% DMSO control for 1 h before 100 μM SA (+) or water (−) treatment for 24 h. YFP fluorescence and chlorophyll autofluorescence are shown in yellow and magenta, respectively. Bar = 20 μm. **b** Protein extracts of *NPR1p:NPR1-YFP* seedlings treated with SA and/or oxicam derivatives as in (**a**) were resolved by SDS-PAGE and analyzed by immunoblotting using anti-NPR1 and anti-PR1 antibodies. CBB staining is shown as a loading control. **c** Protein extracts of Col-0 WT or *npr3-1npr4-3* seedlings treated with SA and/or TNX as in **a** were resolved by SDS-PAGE and analyzed by immunoblotting using anti-NPR1 antibody. CBB staining is shown as a loading control. In **a–c**, experiments were repeated three times with similar results.

did not influence NPR1-GFP migration pattern (Fig. 5e). Moreover, reduction of NPR1-GFP with TCEP did not affect the NPR1-GFP migration pattern even in the NEM pre-treated samples, where the band shift of NPR1-GFP by PEGylation was diminished (Fig. 5f). Accordingly, these results indicate that none of the cysteines in NPR1 form disulfide bonds regardless of SA treatment in vivo. We also tested the redox state of cysteines in soil-grown *35Sp:NPR1-GFP* plants. As previously reported[42,43], the soil-grown *35Sp:NPR1-GFP* plants exhibited severe stunting, development of microlesions, and PR1 accumulation without SA treatment (Supplementary Fig. 10a). We found that NPR1 has no disulfide bond regardless of SA treatment (Supplementary Fig. 10b).

To monitor the TNX effect on the proposed oligomer–monomer transition of NPR1, we next tested an alternative method developed by Mou et al.[21] with minor modification. Total protein was extracted from *35Sp:NPR1-GFP* plants using a HEPES-buffered neutral (pH 7.0) solution without reducing agent and was subjected to SDS-PAGE under either reducing or non-reducing conditions. Without the reducing agent dithiothreitol (DTT), NPR1-GFP was detected at around 250 kDa in untreated control plants, and an additional band of around 90 kDa was present in SA-treated plants (Fig. 6a, top panel), similar to what had been reported previously[21]. The 250 kDa signal was diminished with the addition of DTT, confirming that the 250 kDa and the 90 kDa signals correspond to the disulfide-bridged NPR1 oligomer and monomer, respectively (Fig. 6a, second panel). TNX treatment also induced NPR1 monomer accumulation, and co-treatment with SA resulted in an additive effect (Fig. 6a). This result was contrary to expectations based on the inhibitory effect of TNX on NPR1 levels. Further, there is an apparent discrepancy between the observation of disulfide-bridged NPR1 oligomer (Fig. 6a) and the

results of PEG-Mal labeling assay (Fig. 5e, f). To resolve this discrepancy, we checked whether disulfide-bridged oligomerization of NPR1 occurs in vitro by adding NEM to extraction buffer. Samples from WT and *NPR1p:NPR1-YFP* lines were also included to preclude any saturation effects that might have been due to *35S* promoter-driven *NPR1-GFP* overexpressing plants. Under non-reducing conditions without NEM (−DTT, −NEM), the protein band corresponding to disulfide-bridged NPR1 oligomer was detected in SA-treated *NPR1p:NPR1-YFP* and *35Sp:NPR1-GFP* plants, but not in WT plants, possibly due to lower expression of native NPR1 (Fig. 6b). In contrast, under non-reducing conditions but with the inclusion NEM (−DTT, +NEM), the band corresponding to disulfide-bridged NPR1 oligomer has much weaker signal than in samples without NEM (−DTT, −NEM), whereas the band corresponding to NPR1 monomer has a stronger signal (Fig. 6b). These results suggest that disulfide-bridged oligomerization of NPR1 occurs in vitro under non-reducing conditions. Together these observations indicate that NPR1 oligomer–monomer transitions occur in vitro, but not in vivo. Thus, TNX activity is unlikely to be involved in NPR1 oligomer–monomer transitions.

## Discussion

The *Arabidopsis* suspension cell culture-based screening system identified four structurally distinct types of NSAIDs, all of which potentiate cell death induced by *Pto avrRpm1*. These results may not be surprising as those NSAIDs and the plant immune hormone SA, which potentiates cell death upon induction by avirulent effectors-containing pathogens (Supplementary Fig. 2; refs. [9,26]), uniformly target COX enzymes and show anti-

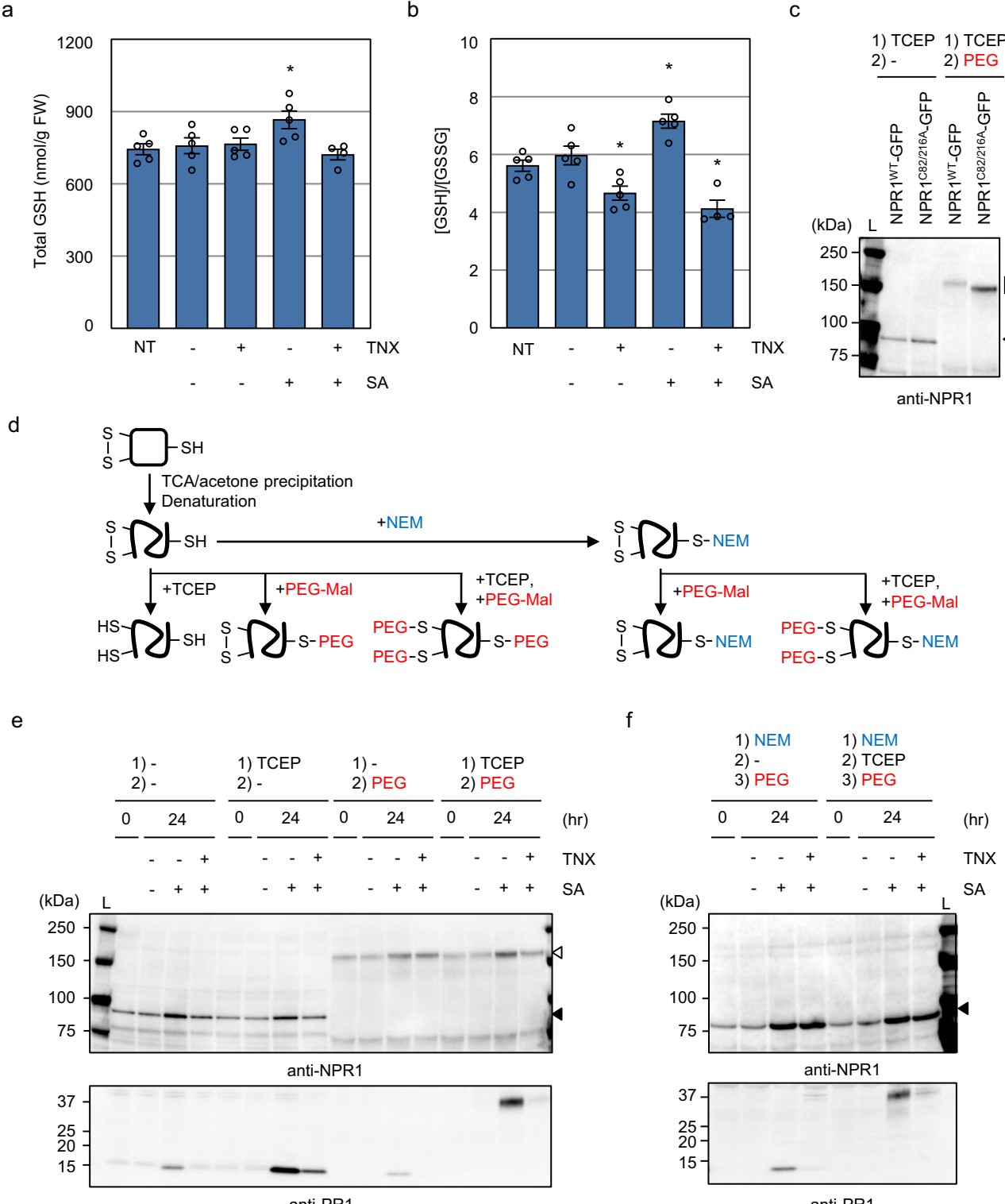

**Fig. 5 TNX inhibits SA-induced glutathione pool redox state changes. a, b** Col-0 WT seedlings were treated with 100 μM TNX (+) or 0.25% DMSO control (−) for 1 h before 100 μM SA (+) or water (−) treatment for 24 h, and total glutathione (**a**) and GSH/GSSG ratio (**b**) were determined. NT represents non-treated controls. Error bars represent means ± SE ($n = 5$ biological replicates for NT, DMSO, TNX, and SA; $n = 4$ biological replicates for SA + TNX). Asterisks indicate significant differences from DMSO control (two-sided Dunnett test, $p < 0.05$). **c** GFP-tagged *Arabidopsis* NPR1^WT and NPR1^C82/216A were expressed in *N. benthamiana* leave. Proteins were extracted and labeled with PEG-Maleimide (PEG). The labeled proteins were resolved by SDS-PAGE and analyzed by immunoblotting using anti-NPR1 antibody. **d** Schematic representation of cysteine labeling workflow. **e, f** *35Sp:NPR1-GFP* seedlings were treated with SA and/or TNX as in **a**. Proteins were extracted and labeled with PEG as described in **d** and "Methods". The labeled proteins were resolved by SDS-PAGE and analyzed by immunoblotting using anti-NPR1 and anti-PR1 antibodies. Black and white arrowheads indicate non-labeled NPR1-GFP (calculated molecular mass: 93 kDa) and PEGylated NPR1-GFP, respectively. L ladder marker. In **a–c**, **e**, **f**, experiments were repeated three times with similar results.

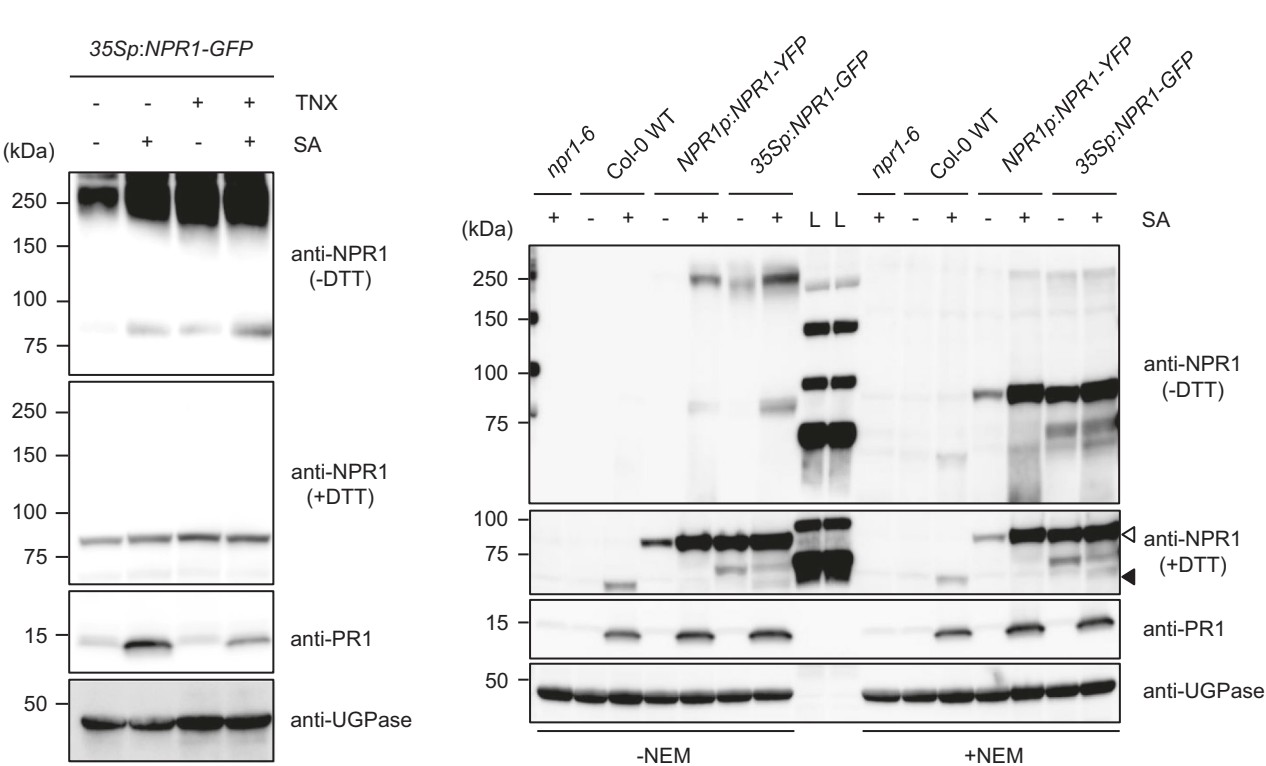

**Fig. 6 Inhibition of NPR1 oligomerization by TCA/acetone precipitation or by the application of a thiol-blocking agent. a** *35Sp:NPR1-GFP* seedlings were treated with 100 µM TNX (+) or 0.5% DMSO control (−) for 1 h before 100 µM SA (+) or water (−) treatment for 24 h. Protein extracts were prepared under reducing agent-free conditions. The proteins were resolved by SDS-PAGE under reducing (+DTT) or non-reducing (−DTT) conditions and analyzed by immunoblotting using anti-NPR1, anti-UGPase, and anti-PR1 antibodies. UGPase levels are shown as a loading control. **b** Seedlings of *npr1-6*, Col-0 WT, *NPR1p:NPR1-YFP*, and *35Sp:NPR1-GFP* were treated with (+) or without (−) 100 µM SA for 24 h. Protein extracts were prepared using a buffer containing (+) or not containing (−) 20 mM *N*-ethylmaleimide (NEM). The proteins were analyzed as in **a**. Black and white arrowheads indicate NPR1 and NPR1-GFP (or NPR1-YFP), respectively. L ladder marker. In **a** and **b**, experiments were repeated three times with similar results.

inflammatory activity in mammalian cells[4]. Although not yet identified, it is possible that SA and NSAIDs bind to a plant protein (or proteins) to potentiate cell death. There are no close homologs of COX in plants, so NSAIDs must act by some different mechanism in plants than in animals[44]. In contrast, Oxicam-type NSAIDs, including TNX, MLX, and PRX, suppress immunity against *Pto*, while others do not, and they inhibit SA signaling. As a possible clue as to their selective effects on inhibiting immunity and SA signaling, oxicams do not contain a carboxy moiety, compared to other NSAIDs. In humans, the carboxy moiety binds to either the channel entrance or the catalytic site of COX[45]. Instead, oxicams indirectly interact with important residues of both the channel entrance and the catalytic site of COX by bridging with two tightly bound water molecules[46]. Thus, in plants, the oxicams and other tested NSAIDs may bind to a common target, but potentially in different conformations that result in distinct phenotypes. Alternatively, oxicams may have yet another plant target, one that selectively regulates the SA-dependent signaling pathway in plants.

Effector-triggered cell death is often associated with immune responses of resistant plants and is potentiated by SA treatment[9], so it is counterintuitive that cell death augmented by oxicams is accompanied by immune suppression. However, similar observations have been reported. For example, the cell death caused by *avrRpm1* is more enhanced in the *Arabidopsis npr1* mutant, which is immune deficient[47]. Similarly, nuclear NPR1 levels were lower at the infection site of an avirulent pathogen causing cell death by

NPR3/4-mediated degradation[14,15]. In contrast, in cells adjacent to tissues infected with avirulent pathogens, accumulated SA promotes formation of cytoplasmic NPR1 condensates that contain regulators of effector-triggered cell death and facilitates its degradation through NPR1-Cullin 3 complex-mediated ubiquitination, and thus promoting cell survival[14]. Thus NPR1 is proposed to function as a negative regulator of effector-triggered cell death[14]. In this respect, the reduction of NPR1 levels as a result of TNX treatment (Fig. 4a, b) is likely the reason for potentiation of effector-triggered cell death accompanied by suppression of immunity. Therefore, we would expect that potentiation of *Pto avrRpm1*-induced cell death by TNX should be lower in the *npr1* mutant compared with WT, although this remains to be tested. In contrast, SA and other NSAIDs, such as IBF and IDM, also potentiate effector-triggered cell death but do not reduce NPR1 levels (Supplementary Fig. 7). Thus, increased effector-triggered cell death is not always associated with low NPR1 levels.

Our transcriptome analysis revealed that TNX broadly represses SA-inducible NPR1-dependent genes (Group I in Fig. 3d, e) but almost half of the genes (Groups II and III in Fig. 3d, e) were not affected. TNX at the concentration used only partially reduces NPR1 protein levels, so it is likely that the level of NPR1 activity that remains under our assay conditions to induce genes that require less NPR1 (Group II). In addition, certain genes are redox-change-sensitive, so that both TNX and SA should induce such genes (Group III). These observations suggest that there is a clear difference in TNX sensitivity within the set of SA-inducible genes.

Because SA has been reported to bind to NPR proteins, we first thought that oxicams directly target NPR proteins to inhibit SA binding. If TNX competitively binds to the SA-binding site of NPR proteins, one would assume that TNX prevents the dissociation of NPR1–NPR4 complexes induced by SA[15,19]. However, TNX did not affect SA-inducible dissociation of the NPR1–NPR4 complex, and did not disrupt NPR1–NPR4 interactions as detected in the yeast two-hybrid assay (Supplementary Fig. 9). We did not detect NPR1–NPR3 interactions in this assay in the presence of SA, unlike what had previously been reported[15]. Therefore, we cannot eliminate the possibility that our yeast two-hybrid conditions were not sensitive enough to assay for TNX effects. Nevertheless, NPR3 and NPR4 are unlikely the targets of TNX, since TNX suppressed SA-induced PR1 accumulation even in *npr3npr4* double-mutant plants (Fig. 4c). This is consistent with findings by Ding et al.[18] who showed that NPR3 and NPR4 function independently of NPR1. Importantly, TNX decreased NPR1 levels and inhibited PR1 accumulation in response to SA in WT and *NPR1p:NPR1-YFP* lines (Fig. 4b, c). In contrast, in *NPR1-GFP*-overexpressing plants, TNX inhibited PR1 accumulation in response to SA without lowering NPR1-GFP levels (Figs. 5e, f and 6a). A large amount of newly synthesized NPR1-GFP, which is produced constitutively in NPR1-GFP overexpressing plants, may mask the TNX effects on NPR1. The mechanism by which oxicams regulates NPR1 levels and its impact on SA signaling remains to be elucidated.

Although TNX may not directly target NPR proteins, we showed that TNX can affect the cellular redox state, which is proposed to be important for NPR1 homeostasis[21,22]. TNX increased oxidation of the cellular redox state as indicated by a decrease in the GSH/GSSG ratio (Fig. 5b). The transcriptome analysis indirectly corroborates this result because oxidative stress-responsive genes were highly expressed in TNX-treated plants (Fig. 3b). In an oxidized cytosolic redox environment modulated by GSH levels, *PR1* expression induced by pathogens or SA is highly reduced[48–50]. It is interesting that both SA and TNX alter cellular redox balance, but the precise relevance between oxidation of cellular redox state by TNX and its inhibitory effect on SA signaling remains to be determined. An important point is that interacting partners of NPR1 are also reported to be regulated in a redox-dependent manner. For example, a TGA-type transcription factor, TGA1, contains a redox-sensitive intramolecular disulfide bond that affects NPR1 interactions[51]. Reduction of the disulfide bond upon SA accumulation stimulated the interaction between NPR1 and TGA1 (ref. [51]). Therefore, it is possible that oxidation of the cellular redox state by TNX most likely disturbs the function of NPR1 indirectly through its interacting transcription factors, resulting in attenuated SA responses.

Although a disulfide-bridged NPR1 oligomer was clearly observed in protein sample preparations under non-reducing neutral conditions (pH 7.0), this oligomerization was mostly abolished following the application of the thiol-blocking reagent (NEM) during sample preparation (Fig. 6a, b). However, we cannot distinguish whether small amounts of disulfide-bridged NPR1 oligomer in NEM-treated samples was formed in vitro (faster than protection by NEM) or in vivo (Fig. 6b). This clear decrease in NPR1 oligomer upon NEM was quite surprising, as it contradicts a previous report[21]. Notably, the denaturation and acidification by TCA also abolished the signal of presumptive NPR1 oligomer (Fig. 5e, f). In addition, the PEG-Mal labeling assay showed that intact NPR1 has no disulfide bonds in vivo, and this is not affected by SA application (Fig. 5e, f). These results indicate that NPR1 can form disulfide-bridged oligomers in vitro, but is highly likely to be the reduced monomer in vivo. This is not unreasonable as the nucleus and cytosol of unchallenged plant

cells are generally maintained in a highly reducing environment that normally disfavors disulfide-bond formation[52]. The apparent inconsistency with previous work might be accounted for by a difference in composition of the extraction buffer. We employed HEPES buffer (pH 7.0) for protein extraction, while Mou et al.[21] used slightly basic Tris buffer (pH 7.5). As the proportion of thiolate anions increases in proportion to the pH of the buffer, the thiols readily form disulfide bonds. In vitro disulfide-bridged oligomerization of NPR1 could therefore proceed more quickly under such conditions[21]. In addition, the slightly basic conditions affect reactivity of NEM toward amines. At pH 7.0, the reaction of the maleimide with thiols proceeds at a rate 1000 times greater than that with amines, but, at pH value above 7.5, a side-reaction toward amines takes place[53]. It is conceivable that the NEM effect is attenuated even by Tris, which contains a primary amine. Furthermore, since the hydrolysis of NEM proceeds faster under basic conditions[42], the instability of NEM in the buffer could be another reason for the discrepancy. However, we cannot exclude the possibility that small amounts of NPR1 oligomerize by forming intermolecular disulfide bonds in vivo, which may be below the detection level in the PEG-Mal labeling assay.

SA treatment was associated with the appearance of monomeric NPR1 under non-reducing condition (in the absence of DTT) as previously described[21] (Fig. 6a), whereas our data suggest that disulfide-bridged oligomerization mainly occurs in vitro during sample preparation. TNX also induced the appearance of the NPR1 monomer band under non-reducing conditions, and the effect was cumulative in the presence of SA, although PR1 accumulation was suppressed by TNX (Fig. 6a). Thus, the increase in signal of an NPR1 monomer band under non-reducing condition does not always correlate with PR1 levels. The underlying mechanism that could explain the appearance of the SA-induced NPR1 monomer band under non-reducing condition remains to be discovered. SA is known to promote interactions between NPR1 and several members of the TGA transcription factor group, or components of the COP9 signalosome complex[33,54]. The interacting partners may shield surface-exposed cysteine residues, resulting in the interference of inter-molecular disulfide-bond formation during sample preparation. Because SA can also promote dephosphorylation of Ser55/Ser59 and sumoylation, as well as subsequent phosphorylation at Ser11/Ser15 of NPR1 (ref. [55]), post-transcriptional modifications of NPR1 may cause conformational changes that influence the reactivity of surface-exposed cysteine residues. SA binding to NPR1 also may induce such changes, as shown in the NPR4–SA interaction[19]. The precise mechanism underlying the appearance of a monomer band under non-reducing conditions needs to be further investigated.

TNX up-regulated the expression of oxidative stress-related genes, thus it may be assumed that TNX directly or indirectly interferes with regulators of cellular redox homeostasis or anti-oxidant enzyme(s). Our transcriptome data revealed that TNX suppresses a subset of SA-inducible NPR1-dependent genes even in *npr1* mutants. This indicates that a potential target of TNX may regulate these genes in an NPR1-independent manner and likely rules out NPR1 as the direct target of TNX. Previously, the cytosolic thioredoxins TRX-h3 and TRX-h5 were identified as regulators of the NPR1 redox state[22]. This work also claims that SAR and SA-induced *PR1* expression are compromised in *trx* mutants due to attenuation of NPR1 monomerization. However, susceptibility to *Pseudomonas* spp. bacteria did not increase either in *trx* mutants or in a cytosolic TRX reductase mutant[22], suggesting that the cytosolic thioredoxin system is unlikely a target of TNX. Instead, it is feasible that TRX proteins affect other processes other than the reduction of NPR1. Target identification will be required to delineate how TNX interferes with SA-dependent

immune signaling in plants. Nonetheless, our study suggests that TNX and other NSAIDs are useful chemical tools for investigating the regulatory mechanisms of SA-dependent signal transduction. In addition, these immunity inhibitors could be used for *Agrobacterium*-based transformation where plant immunity should be suppressed for better efficiency. Thus, in future experiments, it will be important to investigate the TNX immunosuppression effect on plant crop species.

## Methods

**Chemicals**. Two chemical libraries, spectrum collection (1920 chemicals in DMSO (10 mM); MicroSource Discovery Systems Inc.) and NPDepo800 (768 chemicals in DMSO (10 mg/ml); RIKEN Natural Products Depository) were screened for activity. Sodium salicylate (S3007), Tenoxicam (T0909), piroxicam (P5654), meloxicam (M3935), naproxen (N8280), indomethacin (I7378), sulindac (S8139), and mefenamic acid (M4267) were purchased from Sigma-Aldrich. Ibuprofen (098-02641) was purchased from Wako Chemicals.

**Plant materials and growth conditions**. Suspension cultures of *Arabidopsis thaliana* MM1 (Landsberg *erecta* accession) were maintained as described[56]. In brief, *Arabidopsis* MM1 cells were grown in 300 mL conical flasks containing 100 mL Murashige and Skoog (MS) medium supplemented with Gamborg's B5 vitamin, 3% (w/v) sucrose, 0.5 mg/L MES, 0.5 mg/L naphthaleneacetic acid, and 0.05 mg/L 6-benzylaminopurine (pH adjusted to 5.7 with KOH) at 22 °C on a shaker under an 16/8-h light/dark cycle. For experiments using seedlings, *Arabidopsis* seeds were sown on 1/2 MS medium, 0.8% (w/v) agar, and 1% (w/v) sucrose and incubated at 22 °C under an 16/8-h light/dark cycle for 7–10 days. Germinated seedlings were transferred to 1/2 MS liquid medium with 1% (w/v) sucrose and grown for 3 more days before performing experiments. Five- to 6-week-old soil-grown *Arabidopsis* plants (22 °C, under an 8/16-h light/dark cycle) were used for disease resistance assays.

**Cell death assays in *Arabidopsis* suspension cell culture**. *Pto avrRpm1*-induced cell death in *Arabidopsis* suspension culture was monitored as previously described[26]. Briefly, 58.5 μL of *Arabidopsis* suspension cells were dispensed into 96-deep-well plates (260252; Thermo Fisher Scientific), and 0.5 μL of compound was applied to each of two duplicate wells. DMSO and sodium salicylate (20 mM in DMSO) were used as negative and positive controls, respectively. After 1 h incubation, 41 μL of *Pto avrRpm1* suspension (final concentration; OD = 0.2) was applied into one of the duplicate wells. A mock solution without effector was added to the other, control well. After 21 h, cells were stained with 1% Evans blue dye and washed 4× with water. The dye was extracted from cells with 400 μL of 50% methanol, 1% SDS. Absorbance at 595 nm was measured with a microplate reader (iMark Microplate Absorbance Reader; Bio-Rad) to quantify cell death.

**Cell death staining in leaves**. *Pseudomonas syringae* pv. *tomato* DC3000 (*Pto*) *avrRpm1* was grown in King's medium B liquid supplemented with 100 μg/mL rifampicin at 28 °C. *Arabidopsis* seeds were sown as described above and seedlings were grown for a week, then transferred onto rockwool and hydroponically cultivated for 4 weeks under an 8/16-h light/dark cycle. Three days before infection, plants were transferred into small pots containing 100 μM chemicals from the two chemical libraries. A bacterial suspension ($1 \times 10^7$ cfu/mL in 10 mM MgCl$_2$) was injected with a needleless syringe into the abaxial side of leaves, and infected plants were incubated for 6 h. Leaves were stained in lactophenol-trypan blue solution (10 mL of lactic acid, 10 mL of glycerol, 40 mL ethanol, 10 g of phenol, and 8 mg of trypan blue, dissolved in 10 mL of water) and cleared in chloral hydrate (2.5 g chloral hydrate in 1 mL water). Stained cells were observed using an optical microscope (Olympus; BX51). Trypan blue staining area was measured using ImageJ (http://rsb.info.nih.gov/ij/).

**Pathogen growth assay**. *Pseudomonas syringae* pv. *tomato* DC3000 was grown on King's medium B liquid containing 100 μg/mL rifampicin at 28 °C. Five- to six-week-old soil-grown plants grown at 22 °C under an 8/16-h light/dark cycle were syringe-infiltrated with a bacterial suspension of $1 \times 10^5$ cfu/mL in 10 mM MgCl$_2$ and 1% DMSO containing 100 μM oxicam compounds. Bacterial growth in plants was monitored at 3 or 4 days post inoculation.

**RNA extraction and quantitative RT-PCR**. Total RNA samples were extracted using Qiagen RNeasy Plant Mini kit (Qiagen) with DNase I digestion. RNA concentration was quantified with a NanoDrop (Thermo Fisher Scientific). Reverse-transcription was performed with ReverTra Ace qPCR RT master mix (TOYOBO). Quantitative PCR (qPCR) was performed using a Thunderbird SYBR qPCR Mix (TOYOBO) and the analysis was carried out with a real-time thermal cycler (Stratagene Mx3000p; Agilent). Primers for quantitative RT-PCR are as follows: for *PR1*, 5′-TTCTTCCCTCGAAAGCTCAA-3′ and 5′-

AAGGCCCACCAGAGTGTATG-3′; for *PDF1.2*, 5′-TTTGCTGCTTTCGACG-CAC-3′ and 5′-CGCAAACCCCTGACCA TG-3′; *ACT2*, 5′-GATGGCATGAGGAAGAGAGAAAC-3′ and 5′-AGTGGTCGT ACAACCGGTATTGT-3′.

**GUS activity**. Quantitative measurement of GUS activity was performed according to the protocol described by Jefferson et al.[57]. Plant samples were ground with liquid nitrogen and then homogenized in extraction buffer (50 mM NaHPO$_4$, pH 7.0, 10 mM beta-mercaptoethanol, 10 mM EDTA, 0.1% sodium lauryl sarcosine, 0.1% Triton X-100). The homogenate was centrifuged at 20,000$g$ for 10 min at 4 °C. The supernatant was collected and assayed by fluorometric quantitation of 4-methylumbelliferone produced from the glucuronide precursor using a multiplate reader (365-nm excitation filter and 455-nm emission filter; Mithras LB940; Berthold Technologies).

**RNA-sequencing**. Fourteen-day-old seedlings grown on 1/2 MS medium with 0.8% (w/v) agar and 1% (w/v) sucrose were transferred to 1/2 MS liquid medium with 1% (w/v) sucrose and grown for one more day. The following day, SA or SA + TNX was added to the liquid medium (100 μM final concentrations) and the seedlings were incubated for 1 day. RNA-sequencing was performed as described previously[58]. Purified double-stranded cDNAs were subjected to Covaris shearing (parameters: intensity, 5; duty cycle, 20%; cycles/burst, 200; duration, 60 s). The libraries were sequenced on an Illumina Hiseq 2500. The sequence data have been deposited in the NCBI Gene Expression Omnibus (GEO) database and are accessible through GEO Series accession number GSE154593. Sequence reads to gene associations were carried out using the considerations described previously[58]. Quality-filtered libraries were aligned to *Arabidopsis* TAIR10 genome using Bowtie version 0.12.8 (ref. [59]). Unaligned reads from the previous step were aligned to transcript sequences of *Arabidopsis* Col-0 (ftp://ftp.Arabidopsis.org/home/tair/Sequences/blast_datasets/TAIR10_blastsets/TAIR10_cdna_20101214_updated) using Bowtie version 0.12.8 (ref. [59]). The uniquely aligned reads were used for downstream analysis. Differential expression analysis was performed using the R statistical language version 2.11.1 with the Bioconductor[60] package, edgeR version 1.6.15 (ref. [61]) with the exact negative binomial test using tagwise dispersions. Gene ontology analysis was performed using DAVID Bioinformatics Resources 6.8 (http://david.ncifcrf.gov/home.jsp)[62].

**Microscopy**. Subcellular localization of NPR1-YFP was observed in cotyledon epidermal cells using an inverted laser scanning confocal microscope (LSM700; ZEISS). NPR1-YFP was excited at 488 nm, and the emission was collected using a BP 490–555 nm filter. Autofluorescence from chlorophyll was excited at 555 nm and the emission was collected using a LP 640 nm filter. Obtained images were processed using ZEN 2.1 SP1 software (ZEISS) and Photoshop CC 2019 (Adobe).

**Generation of anti-NPR1 antibody**. Antigen was prepared with reference to the method described by Mou et al.[21] A truncated cDNA clone of *NPR1* (*tNPR1*, 1-1395 bp) was inserted into the *Nde* I/*Not* I sites of the pET-26 vector (Novagen) in-frame with C-terminal His tag. The pET-26-tNPR1-His plasmid was then transformed into Rosetta (DE3) pLysS (Novagen). The overnight culture at 37 °C was transferred to fresh LB medium containing 50 μg/mL kanamycin and was incubated to OD$_{600}$ = 1.0 at 37 °C. Protein synthesis was induced with isopropyl β-D-thiogalactopyranoside at a final concentration of 0.2 mM. The induced cells were incubated overnight at 16 °C. The *E. coli*-expressed tNPR1-His protein was extracted and purified using TALON Metal Affinity Resin (Clontech). For rabbit immunization, the tNPR1-His proteins were further purified by separation on SDS-PAGE. Polyclonal antisera were raised in rabbits (Operon Biotechnologies). Anti-NPR1 antibodies were purified from antisera by affinity chromatography using a tNPR1-His-immobilized column (Hitrap NHS-activated HP; GE Healthcare). To prepare for the tNPR1-His-immobilized column, tNPR1-His proteins were further purified by HiLoad 16/600 Superdex 200 (GE Healthcare) with 200 mM NaHCO$_3$, pH 8.3, 500 mM NaCl, 2 mM DTT, 0.1% CHAPS as a mobile phase.

**Protein extraction and immunoblotting**. Plant samples were ground with liquid nitrogen and then homogenized in extraction buffer (20 mM HEPES-NaOH, pH 7.0, 150 mM NaCl, 2 mM EDTA, 1% Nonidet P-40, 0.5% sodium deoxycholate, 2 mM DTT, 100 μM MG115, and 1× cOmplete™ EDTA-free Protease Inhibitor Cocktail (Roche)). For protein extraction under non-reducing conditions, DTT was omitted from this extraction buffer. For protein extraction under conditions where the maintenance of the thiol group redox state was required, 20 mM *N*-ethylmaleimide (NEM; E0136; Tokyo Chemical Industry Co.) was added to the extraction buffer instead of DTT. The homogenate was centrifuged at 20,000$g$ for 10 min at 4 °C, and the supernatant was collected and subjected to SDS-PAGE. Proteins were then electroblotted onto a PVDF membrane using a semidry blotter (trans-blot turbo transfer system; Bio-Rad). Membranes were blocked overnight at 4 °C in TBS-T (50 mM Tris-HCl, pH 7.5, 150 mM NaCl, and 0.05% Tween 20) with 5% skim milk. Membranes were then incubated with either anti-UGPase (1:5000; AS05 086; Agrisera), anti-PR1 (1:5000; AS10 687; Agrisera), anti-HA (1:5000;

12013819001; Roche), anti-Myc (1:1000; sc-789; Santa Cruz Biotechnology), or anti-NPR1 antibodies (1:2000) diluted with TBS-T with 5% skim milk at room temperature for 1 h. After washing with TBS-T, the membranes were incubated with horseradish peroxidase-conjugated anti-rabbit IgG (1:10,000; NA934; GE Healthcare) diluted with TBS-T for 1 h at room temperature. Bound antibodies were visualized using SuperSignal West Dura Extended Duration Substrate (Thermo Fisher Scientific) and bands were imaged using an image analyzer (ImageQuant LAS 4000 imager; GE Healthcare).

**Yeast two-hybrid assay.** Yeast strain AH109 transformed with pGADT7-NPR3/4 and strain Y187 transformed with pGBKT7-NPR1 were kindly provided by Dr. Xinnian Dong (Duke University). The transformed yeast strains were streaked on synthetic dropout (SD) media lacking Leu and Trp, respectively. For yeast mating, single colonies were grown overnight in YPDA liquid media. The cultures were then streaked on SD media lacking Leu and Trp (−LT). Single colonies were grown in SD (−LT) liquid media for 1 day and spotted (1:100 dilution) on SD lacking Leu, Trp, and His (−LTH) supplemented with 3 mM 3-aminotriazole, with or without 100 μM SA and/or 100 μM TNX and incubated at 30 °C to monitor the interaction between bait and prey.

**Transient gene expression in *Nicotiana benthamiana*.** The CDS coding region of *NPR1* was amplified using primers (5′-CATTTACAATTATCGATATGGACACC ACCATTGATGGATTC-3′, 5′-TGCTCACCATGGATCCCCGACGACGATGAG AGAGTTTAC-3′) by PCR with KOD One PCR master mix (TOYOBO). The amplified fragment was cloned in-frame with c-terminal eGFP and placed under the control of the CaMV 35S promoter of epiGreenB binary vector[63]. Site-directed mutagenesis was performed using In-Fusion cloning method according to the manufacturer's instruction (In-Fusion HD cloning Kit; Takara Bio). *Agrobacterium tumefaciens* C58C1 strains carrying the binary vectors were grown in LB liquid medium overnight. Cultures were centrifuged, and the pellets were resuspended in infiltration buffer (10 mM MES-NaOH, pH 5.6, 10 mM $MgCl_2$, and 100 μM acetosyringone). Infiltration of *Agrobacterium* was done at $OD_{600} =$ 0.3 and protein was extracted 2 days after infection.

**TCA/acetone precipitation and protein labeling.** Plant samples were ground with liquid nitrogen and then homogenized in 10% TCA in acetone. The homogenate was incubated at −20 °C for 1 h and centrifuged at 20,000g for 10 min at 4 °C. The supernatant was discarded and the pellet was washed two times with ice-cold acetone. The protein pellet was homogenized and incubated in homogenizing buffer (50 mM HEPES, pH 7.0 and 2% SDS) containing 10 mM 2k PEG-maleimide (PEG-Mal; SUNBRIGHT ME-020MA; NOF) or 40 mM tris (2-carboxyethyl) phosphine (TCEP; 203-20153; FUJIFILM Wako Pure Chemicals) for 30 or 90 min, respectively. The homogenate was then centrifuged at 20,000g for 10 min, and the supernatant was collected. Half of the TCEP-treated sample was precipitated with chloroform/methanol method[64]. The precipitated protein was washed twice with ice-cold acetone and labeled with PEG-Mal as described above. For pre-treatment of NEM, the pellet was homogenized and incubated in homogenizing buffer containing 40 mM NEM for 90 min. The homogenate was then centrifuged at 20,000g for 10 min, and the supernatant was collected. The NEM-treated sample was precipitated with chloroform/methanol method to remove NEM, and the precipitated protein was labeled with PEG-Mal as described above. The protein sample was mixed with 2× SDS-PAGE sample buffer without reducing agent and subjected to immunoblot analysis.

**GSH measurement.** GSH measurements were performed as described previously[65]. Plant samples were ground with liquid nitrogen and then homogenized in 6% $HClO_4$. The homogenate was centrifuged at 20,000g for 10 min at 4 °C. The supernatant was collected and neutralized with 1.25 M $K_2CO_3$, and then centrifuged at 20,000g for 10 min at 4 °C. The supernatant was divided in two to measure total GSH (reduced GSH + 2× GSSG) and GSSG. For total GSH measurements, the sample was incubated in assay buffer consisting of 100 mM potassium phosphate buffer, pH 7.5, 5 mM EDTA, 0.6 mM 5,5′-dithiobis (2-nitrobenzoic acid), 0.2 mM NADPH, and 0.5 U/mL glutathione reductase from yeast (46541005; Oriental yeast). The reaction rate was monitored by a microplate reader (Infinite F200; Tecan) equipped with a 405 nm (bandwidth 10 nm) filter. For GSSG measurements, the sample was immediately mixed with thiol-masking reagent 2-vinylpyridine (20 mM final concentration) after the sample separation and incubated for 1 h at 25 °C, and then measured by the same method as total GSH. The concentrations of total GSH and GSSG were calculated using GSH or GSSG standard curves simultaneously in the assay.

**Reporting summary.** Further information on research design is available in the Nature Research Reporting Summary linked to this article.

## Data availability

RNA-seq data from this study have been deposited in the NCBI Gene Expression Omnibus (GEO) and are accessible through GEO series accession number GSE154593. Source data are provided with this paper.

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

## Acknowledgements

We thank Dr. Xinnian Dong, Dr. Xin Li, Dr. Mary Wildermuth, Dr. Bethany Huot, and Dr. Yoshihiro Narusaka for sharing materials. We also thank all the members of Ken Shirasu laboratory for fruitful discussions; Dr. Yasuhiro Kadota and Dr. Anuphon Laohavisit for critically reading the manuscripts; and Mrs. Satoko Morino for lab help. This work was supported in part by Grant-in-Aid for Scientific Research (KAKENHI) (15H05959, 17H06172, 20H05909, 19039034, and 24228008 to K.S., 15K18651, 17K07679, and 20H02995 to S.A.).

## Author contributions

N.I. and K.S. conceived the project. N.I., S.-w.C., Y.N., I.S., S.A., and K.T. conducted experiments and analyzed results. N.I., S.-w.C., Y.N., I.S., and K.S. conceived and designed experiments. H.O. provided the RIKEN Natural Product Depository compounds. S.Y.H. provided *NPR1p:NPR1-GFP* plants prior to publication. N.I., S.-w.C., Y.N., S.A., S.Y.H., and K.S. wrote the paper. All authors reviewed the manuscript.

## Competing interests

The authors declare no competing interests.
