## [Peer Review File · Nature Communications]

Oxicam-type nonsteroidal anti-inflammatory drugs inhibit
NPR1-mediated salicylic acid pathwayREVIEWER COMMENTS

Reviewer #1 (Remarks to the Author):

The manuscript by Ishihama et al. is aimed at understanding regulation of plant immunity by nonsteroidal anti-inflammatory (NSAID) compounds. It is proposed that the compounds tested are structural analogues of the phytohormone salicylic acid (SA). In plants, SA rapidly accumulates in response to infection by biotrophic pathogens to orchestrate immunity. SA induces marked changes in the transcriptome, a process largely mediated by the immune-transcriptional regulator NPR1. Conversely, adapted pathogens deliver effector molecules into plant cells to suppress host immunity. Recognition of effector molecules by plant cells often triggers programmed cell death, which is facilitated by SA and negatively regulated by NPR1. Previously, the authors identified several NSAID as potentiators of plant cell death in response to the avirulent pathogen *Pto avrRPM1*. Here, the authors build on their previous work and aim to elucidate the molecular mechanisms by which NSAID, particularly TNX, impact immunity.

While the manuscript uncovers an interesting effect of TNX on SA/NPR1 signalling, unfortunately the data create more questions than answers. The findings do not provide an understanding for why TNX (and other oxicam-type NSAIDs) potentiate effector-triggered immunity but suppress SA-mediated immunity. Moreover, the data do not explain why TNX suppresses only a subset of SA/NPR1-induced genes. While this suggest two different mechanisms are at play, the manuscript does not reveal new mechanistic insight into SA/NPR1 signalling. Thus, the authors have uncovered a potentially interesting new tool in TNX to dissect SA/NPR1 signalling further but unfortunately have not done so in this manuscript. The manuscript's findings are therefore far too preliminary in nature to expose new aspects of immune signalling and instead lead to a lengthy and highly speculative Discussion.

Specific comments:

(1) Fig1A: attempts to show that NSAID compounds facilitate cell death in response to *Pto avrRpm1*. As noted by the authors, most of the compounds shows cytotoxic effects on concentrations above 25uM. This data seems to indicate an additive effect of programmed cell death in response to *avrRpm1* and cytotoxicity of the compounds tested rather than enhanced cell death in response to *avrRpm1*.

(2) Fig 1c is not terribly convincing. and would be better if quantified (from images or by ion leakage). Moreover, it should contain non-oxicams compounds to justify the statement that specifically oxicam-type NSAIDs potentiate effector-triggered cell death.

Also, in this assay plants were treated with the compounds 3 days before pathogen inoculation, whereas in Fig 1b plants were concomitantly infiltrated with the indicated compounds and Pto. What is the reason for the different protocols?

(3) Line 123 is an overstatement. Oxicam-type NSAIDs have some similar chemical features to SA but they are not structurally similar.

(4) Fig 2d: As shown this figure seems to display 3 identical treatments (-SA).

(5) From the text and legend it is unclear what Fig 3a shows and compared: plants treated with TNX alone (vs control) or with SA+TNX (vs SA alone)? I assumed it is the first but ambiguity should be removed in the text.

(6) The authors focus solely on the effect of TNX on SA-induced genes (Fig 3c-d). However, SA also downregulates a similar size set of genes, so what is the effect of TNX on these SA-suppressed genes?

(7) On page 6 the text several times states that gene expression was repressed/decreased “to less than 0.5-fold”. It is unclear what this means. Fold change in comparison to what?

(8) From Fig 4c it is concluded that TNX suppressed NPR1 levels independent of NPR3/4, but are SA-induced NPR1 protein levels at all dependent on NPR3/4 in this assay? If they were, one would expect that the *npr3/4* mutant accumulates more NPR1 than the WT in response to SA treatment. Perhaps the timing of sampling is somewhat late here?

(9) Lines 231-233 state that NPR1 was denatured with TCA/acetone to prevent further in vitro oxidation? Denaturing the protein does not prevent further oxidation of Cys residues (in fact, it could promote it!). To prevent oxidation an alkylating agent should be used, so this could be a major problem in the assay of Fig 5c.

(10) Fig 5c presents several further problems:

First, NPR1-GFP protein levels are unresponsive to TNX treatment, which is at odds with findings presented in Fig 4. The text puts this down to a saturating effect of the 35S promoter but this is unlikely at the level of protein loaded on to the blots.

Second, the authors conclude that non of the Cys residues in NPR1-GFP form disulfide bonds. This is at odds with the literature and the data presented here also do not necessarily support this. PEG-Mal induces only a shift of ~2kDa per Cys residue, so the provided blots do not have the resolution to

conclude this, especially if only a few Cys residues are involved in disulfide bonding. Moreover, because PEG-Mal labelled NPR1-GFP already runs at an unexpected higher molecular weight, this is likely not a suitable assay to determine Cys oxidation of NPR1-GFP. Hence the assay is inconclusive.

(11) Lines 262-264: the described data that DTT diminished the 250kDa signal is not shown in Fig 6a or anywhere else.

(12) Fig 6a The fact that TNX induces NPR1 monomer either on its own or in combination with SA creates more questions than answers. This does not square up with the effect of TNX on cellular redox.

(13) The data in Fig 6c are quite surprising, as a similar experiment has been reported by Mou et al. (2003) who showed that NEM and IOD did not affect NPR1-GFP oligomer detection. The effect of NEM in this figure is particularly surprising as the Methods indicate that the protein sample was not incubated and kept at 4C. NEM is not very reactive at low temperatures and requires at least incubation at room temperature or higher to block Cys residues.

Nonetheless, the authors observe an oligomeric signal in the +NEM samples (albeit lower than in the - NEM samples), so it seems premature to conclude that NPR1 oligomerisation occurs only in vitro and not in vivo.

Other editorial comments:

(1) Page 2 (top): In the description of targets of SA in plants and its derivatives in mammals, primary references should be used instead of reviews. Also, SA accumulates in plants cells upon recognition of biotrophic pathogens and may not be as effective against necrotrophics. The text should be revised accordingly.

(2) Page 3 (top): "NPR1 was also reported to serve as an intrinsic link between daily redox rhythms and the circadian clock (20). Thus, NPR1 has been proposed to be a redox sensor in plants." It is not clear the relevance of this information here.

(3) Page 3 (bottom): Introduction of key terminology for this study, such as basal immunity, hypersensitive response and effector triggered immunity, would help the non-expert reader.

Reviewer #2 (Remarks to the Author):

The manuscript is a result of a research concept aiming at identifying chemicals that prime the plant immune response. Here, different nonsteroidal anti-inflammatory drugs (NSAIDs) were tested. Although the compound TNX primed the cell death response, it suppressed a big sector of the NPR1-dependent response towards salicylic acid. The authors went on to characterize the mechanism how TNX might interfere with the function of SA-activated NPR1 and by doing so they got contradictory results to a longstanding working model for NPR1 action.

It is well established that SA-mediated responses correlate with the activation of the anti-oxidative system, leading to higher glutathione levels and to a higher GSH/GSSG ratio. According to the current model, this redox shift leads to the reduction of intermolecular disulfide bridges in the cytosolic multimeric NPR1 complex, allowing nuclear translocation of resulting monomers. In this manuscript, it is shown, that TNX interferes with the establishment of the „reduced state“ through an unknown mechanism. Therefore, it was expected that TNX would interfere with the reduction of the NPR1 multimer to monomers.

However, TNX enforces monomer formation and acts additively with SA in this respect (Figure 6a). When preparing the extract in the presence of 10% TCA, which interferes with the oxidation of thiol groups, the oligomer is not formed any more. This result suggests that disulfide-bridged NPR1 oligomers are not present *in vivo* but rather form only *in vitro* when keeping the protein in a buffer without DTT. Pre-treatment of plants with SA or TNX would change the redox state of interaction partners of NPR1, which would protect a fraction of NPR1 against oxidation. If this were true, it would question the established model of the redox-regulation of NPR1. Figure 6c supports this conclusion: NEM treatment of plant extracts, which protects reduced thiol groups against oxidation, prevents – like DTT treatment – oligomerisation of NPR1. The third piece of evidence comes from another alkylation experiment with TCA-precipitated proteins. Here mPEG was used which adds a 2kDa moiety to each cysteine. No influence of either SA or TNX or DTT on the amount of added mPEG was observed. All these experiments point at the situation that *in vivo*, NPR1 seems to be in the reduced state all the time. Oligomerisation through oxidation might be an *in vitro* artefact and previous researchers might have been misled. To my opinion, this is the most important lesson of the manuscript. Since it would generate a little „earth quake“ in the community, experiments should not leave any doubt.

Fig. 5: Criticism on the mPEG experiment might be raised since it is not shown whether a difference of 2 or 4 kDa, which would be indicative for the oxidation of 1 or 2 residues, respectively, can be seen with this method. It would be good to perform such an experiment side by side with a construct which contains 17 rather than 19 cysteines. Maybe this can be done with recombinant protein. Moreover, the experiment should be repeated in a buffer of pH 8. When performing the reaction at pH 7, the authors cannot guarantee that all residues can be alkylated. In the worst case, the mobility of the alkylated proteins would always be the same since the potentially oxidized cysteine cannot be alkylated even

after reduction because of the wrong pH of the buffer. It is unlikely, but it has to be done. Moreover, it has to be taken into account, that SA treatment does not completely monomerize the oligomer. Thus, it is expected that only a small portion of the protein can be alkylated to a higher

degree in extracts from SA treated plants. Maybe this is below the detection level.

Since the experiment has to be repeated anyhow, I recommend to use –DTT control at the same time. It would thus integrate Fig. 6b showing that –DTT does not change the mobility of NPR1. The subsequent labelling experiments with mPEG with PR1 used as a control show that TCA precipitation does not reduce an oxidized protein in the absence of DTT.

Minor points concerning this figure: Can the authors comment why the band for PR1 in untreated plants only shows up in the DTT-treated extracts? It would be nice to point out that the 35S-NPR1-GFP construct is in the *npr1-1* background.

However, these experiments were done with seedlings grown in sucrose containing medium. This might have been necessary due to the drug treatment, but the experiments addressing the effect of SA on the redox state of NPR1 should also be done with soil-grown plants.

If these experiments continue to point to the direction that NPR1 does not form disulfide bridged oligomers, this very important message should be mentioned in the abstract or even in the title. Many courses at universities all over the world teach the students how NPR1 works and part of it seems to be based on the misinterpretation of experiments.

There is another discrepancy to the current model as proposed by the Dong group. In the absence of SA, NPR4 degrades NPR1. In the presence of SA, the interaction of NPR1 and NPR4 is disrupted, so that NPR1 can accumulate. If there is even more SA, NPR3 binds SA and initiates degradation of NPR1, which would otherwise interfere with cell death. If this model were true, the *npr4* mutant and the *npr3 npr4* double mutant should show elevated NPR1 levels in the absence of SA, which was the case in Fu et al Nature 2012. This was not reproduced in Figure 4 of this manuscript and this should be pointed out. Since TNX interferes with NPR1 accumulation. One hypothesis was that TNX would maintain the interaction between NPR4 and NPR1 even in the presence of SA. This was excluded.

Other points.

Concerning the TNX part of the story, I have the following suggestions. Since the effect of TNX on NPR1-mediated immune responses is in the focus of the paper, Figure 1a, which addresses the priming effect of different drugs on PTO-mediated cell death, should go to the supplement.

When explaining Figure 3, it should be pointed out that TNX has negative effects on NPR1-dependent gene expression even in the absence of NPR1. This seems to allow the conclusion, that other processes that happen in the absence of NPR1 are affected.

Discussion: Line 423: I had the impression that Tada et al. claim that SAR is compromised in *trx* mutants, also PR1 expression is compromised. Still, it is feasible that TRXs are required for other processes and not for the reduction of NPR1.

Dear editor and reviewers

We would like to thank the referees for their thoughtful comments, invaluable feedback as well as constructive criticisms, which we believe have greatly improved our manuscript. Especially, we improved the labeling method to determine the redox state of cysteine residues in NPR1 and performed additional experiments to address the reviewers' concerns. In addition, we have provided a point-by-point reply to address specific concerns that each referee had, and we addressed the points raised. We sincerely hope that you and the referees will find our revised manuscript acceptable for publication.

On behalf of all the authors,

Nobuaki Ishihama and Ken Shirasu

REVIEWER COMMENTS

Reviewer #1 (Remarks to the Author):

The manuscript by Ishihama et al. is aimed at understanding regulation of plant immunity by nonsteroidal anti-inflammatory (NSAID) compounds. It is proposed that the compounds tested are structural analogues of the phytohormone salicylic acid (SA). In plants, SA rapidly accumulates in response to infection by biotrophic pathogens to orchestrate immunity. SA induces marked changes in the transcriptome, a process largely mediated by the immune-transcriptional regulator NPR1. Conversely, adapted pathogens deliver effector molecules into plant cells to suppress host immunity. Recognition of effector molecules by plant cells often triggers programmed cell death, which is facilitated by SA and negatively regulated by NPR1. Previously, the authors identified several NSAID as potentiators of plant cell death in response to the avirulent pathogen *Pto avrRPM1*. Here, the authors build on their previous work and aim to elucidate the molecular mechanisms by which NSAID, particularly TNX, impact immunity.

While the manuscript uncovers an interesting effect of TNX on SA/NPR1 signalling, unfortunately the data create more questions than answers. The findings do not provide an understanding for why TNX (and other oxicam-type NSAIDs) potentiate effector-triggered immunity but suppress SA-mediated immunity. Moreover, the data do not explain why TNX suppresses only a subset of SA/NPR1-induced genes. While this suggest two different mechanisms are at play, the manuscript does not reveal new mechanistic insight into SA/NPR1 signalling. Thus, the authors have uncovered a potentially interesting new tool in TNX to dissect SA/NPR1 signalling further but unfortunately have not done so in this manuscript. The manuscript's findings are therefore far too preliminary in

nature to expose new aspects of immune signalling and instead lead to a lengthy and highly speculative Discussion.

Specific comments:

(1) Fig1A: attempts to show that NSAID compounds facilitate cell death in response to *Pto avrRpm1*. As noted by the authors, most of the compounds shows cytotoxic effects on concentrations above 25uM. This data seems to indicate an additive effect of programmed cell death in response to *avrRpm1* and cytotoxicity of the compounds tested rather than enhanced cell death in response to *avrRpm1*.

> Thank you for this comment. As is often the case in chemical biology, many compounds, especially synthetic ones, cause side effects at higher concentrations. Toxic levels of chemicals are often higher in cell cultures or liquid-grown plants than soil-grown plants, probably due to different mobilities, stabilities, or permeabilities depending on how you apply the chemicals. Therefore, for each experimental system, the toxic levels of each compound need to be clarified for better interpretation. For this purpose, we previously showed that the toxic levels of each compound in cell cultures in the original Fig. 1a. We have now moved this figure to the supplemental information as the reviewer 2 suggested (see below). We believe that the effects of NSAIDs are not additive at lower concentrations. For example, as shown in Supplemental Fig. 2 (formerly Fig. 1a), 12.5uM of TNX alone has no effect on the cell culture, although it doubles cell death levels when inoculated with *Pto*. Similarly, the sum of 'increases in cell death rate' caused by each NSAIDs at lower concentration and *Pto avrRpm1* alone was lower than that caused by co-treatment with the NSAIDs and *Pto avrRpm1*.

(2) Fig 1c is not terribly convincing. and would be better if quantified (from images or by ion leakage).

> Thank you for your comments. We reperformed the experiment of the original Fig. 1c and quantified the stained area. The obtained data were incorporated into the new Fig. 1b, c.

Moreover, it should contain non-oxicams compounds to justify the statement that specifically oxicom-type NSAIDs potentiate effector-triggered cell death.

> We never stated "specifically oxicom-type NSAIDs potentiate effector-triggered cell death". On

the contrary, we stated “The *Arabidopsis* suspension cell culture-based screening system identified four structurally distinct types of NSAIDs, all of which potentiate cell death induced by *Pto avrRpm1*” in the Discussion. Similarly, in the result section, we clearly stated “*Pto avrRpm1*-induced cell death was significantly enhanced in a dose-dependent manner by the application of propionic acid derivatives (IBF and naproxen (NPX)), acetic acid derivatives (indomethacin (IDM) and sulindac (SLD)), anthranilic acid derivatives (mefenamic acid (MFA)), and oxicam derivatives (TNX, meloxicam (MLX) and piroxicam (PRX))”. To clarify our emphasis on oxicam, we have edited the main text to “We conclude that, while both oxicam-type and non-oxicam-type NSAIDs potentiate effector-triggered cell death in suspension cell cultures, only oxicam-type NSAIDs suppressed host immunity in *Arabidopsis* plants” on line 131 in the manuscript.

Also, in this assay plants were treated with the compounds 3 days before pathogen inoculation, whereas in Fig 1b plants were concomitantly infiltrated with the indicated compounds and *Pto*. What is the reason for the different protocols?

> Thank you for the comments. This is due to the timing of phenotype scoring. In pathogen (*Pto* DC3000) inoculation assay, we observed bacterial growth three- or four-days post inoculation, while *AvrRpm1*-triggered cell death is a faster process, often proceed within hours. Thus, chemicals were taken up by roots prior to the inoculation with *Pst avrRpm1* to look at cell death response (Fig. 1b, c).

(3) Line 123 is an overstatement. Oxicam-type NSAIDs have some similar chemical features to SA but they are not structurally similar.

> Thank you for your comments. We agree that “similar” is a vague word, and we have revised the statement as follows: Given that the oxicam-type NSAIDs and SA share a common target in animal cells, ...

(4) Fig 2d: As shown this figure seems to display 3 identical treatments (-SA).

> Thank you for pointing this out. The charts should have labeled with not only “-SA” but also “+TNX”. This is our mistake on labeling. We have now labeled properly.

(5) From the text and legend it is unclear what Fig 3a shows and compared: plants treated with TNX alone (vs control) or with SA+TNX (vs SA alone)? I assumed it is the first but ambiguity should be removed in the text.

> Thank you for this comment. To eliminate ambiguous description, we have revised the statement as follows: We compared transcriptome profiles between Col-0 WT and *npr1-1* seedlings treated with either i) mock, ii) 100 μ M SA, iii) 100 μ M TNX, or iv) 100 μ M SA + 100 μ M TNX (Supplementary Data 1). Comparison between TNX-treated Col-0 WT and mock-treated Col-0 WT showed that 2,402 genes were induced (FDR = 0.001 and \log_2 fold change (\log_2 FC) > 1) and 2,662 were repressed (FDR = 0.001 and \log_2 FC < -1) by TNX treatment (Fig. 3a and Supplementary Data 2 and 3).

(6) The authors focus solely on the effect of TNX on SA-induced genes (Fig 3c-d). However, SA also downregulates a similar size set of genes, so what is the effect of TNX on these SA-suppressed genes?

> Thank you for the constructive comment. We re-analyzed our transcriptome data and showed the effect of TNX on SA-suppressed genes in Supplemental Fig. 5.

(7) On page 6 the text several times states that gene expression was repressed/decreased “to less than 0.5-fold”. It is unclear what this means. Fold change in comparison to what?

> Thank you for this comment. We have thoroughly revised this section of the manuscript (line 174-193). Accordingly, we also described how each comparison was made in our gene expression analysis.

(8) From Fig 4c it is concluded that TNX suppressed NPR1 levels independent of NPR3/4, but are SA-induced NPR1 protein levels at all dependent on NPR3/4 in this assay? If they were, one would expect that the *npr3/4* mutant accumulates more NPR1 than the WT in response to SA treatment. Perhaps the timing of sampling is somewhat late here?

> Thank you for this comment. Indeed, we did not observe higher NPR1 accumulation in the *npr3npr4* mutant compared to Col-0 WT at 24 hr after SA treatment (Fig. 4c). This is consistent with the previous studies, as Fu et al. showed that NPR1 levels are higher in the *npr3npr4* mutant than Col-0 WT at 4 hr after SA treatment, but not at 8 hr. Thus, it is likely that later time points (i.e. 8 hr~24 hr), there is no difference between *npr3npr4* and Col-0 WT.

(9) Lines 231-233 state that NPR1 was denatured with TCA/acetone to prevent further in vitro oxidation? Denaturing the protein does not prevent further oxidation of Cys residues (in fact, it could promote it!). To prevent oxidation an alkylating agent should be used, so this could be a

major problem in the assay of Fig 5c.

> We acknowledge reviewer #1's opinion that "denaturation" per se has a potential to promote oxidation of Cys residues because denaturation can expose thiols that are buried and are protected from oxidation in the native protein structures. However, the TCA/acetone method has the effect of not only denaturation but also acidification, which effectively protects free thiols from oxidation as described below (Le Moan et al., (2008) *Methods Mol Biol*, 476, Hansen et al., (2009) *PNAS*, 106, 422). Firstly, acidification by TCA prevents the thiols from deprotonating into a thiolate anion, which is considerably more powerful as a nucleophilic agent, thus protecting the thiols from redox-related modifications. Secondly, denaturation can decrease the reactivity of redox-sensitive thiols in proteins. Free cysteine has a pKa of 8.3, while redox-sensitive proteins often contain lower-pKa thiols, which predominantly exist as thiolate anion even at physiological pH, depending on their microenvironment in a protein. After denaturation, these thiols do not show higher reactivity even in the subsequent processes due to the collapse of the microenvironment. Because alkylated thiol groups no longer react with maleimide, alkylating agents are not applicable for PEG-Mal labeling assays.

(10) Fig 5c presents several further problems:

First, NPR1-GFP protein levels are unresponsive to TNX treatment, which is at odds with findings presented in Fig 4. The text puts this down to a saturating effect of the 35S promoter but this is unlikely at the level of protein loaded on to the blots.

> Thank you for your comment. We believe that the difference is due to the promoters. In Fig. 4, the NPR1 native promoter was used whereas Fig. 5 uses 35S. We think that an extensive amount of newly synthesized NPR1-GFP driven by the 35S promoter may mask the TNX effects on NPR1 (see in Discussion). In order not to cause misinterpretation of the word "saturation" as a gel artifact, we have eliminated the statement "The absence of any apparent effects on NPR1 levels is likely due to saturation effects in 35S promoter-driven *NPR1-GFP*-overexpressing plants." from the Result.

Second, the authors conclude that non of the Cys residues in NPR1-GFP form disulfide bonds. This is at odds with the literature and the data presented here also do not necessarily support this. PEG-Mal induces only a shift of ~2kDa per Cys residue, so the provided blots do not have the resolution to conclude this, especially if only a few Cys residues are involved in disulfide bonding.

> We acknowledge reviewer #1's opinion that the resolution of PEG-Mal labeling assay is critical in our study. We initially believed that the resolution was enough but here we decided to verify this by

conducting new experiments (See also the reviewer 2's comment). Firstly, we have improved our PEG-Mal labeling assay so that all free cysteines can be labeled while maintaining specificity. In particular, we optimized the labeling reaction and used TCEP instead of DTT for the reduction of disulfide bonds, because TCEP is more potent than DTT under pH 7.0 condition (Han and Han (1994) *Anal Biochem* 220, 5). Please note that the control PR1 contains 6 Cys residues making 3 disulfide bonds but no free Cys residues. The molecular weight of PR1 is ~14.8 kDa and when PEGylated, it runs ~37 kDa (in the old Fig. 5c and new Fig. 5e, f). This means 6xPEG-Mal runs about 22 kDa equivalent for PR1 labeling. Thus, the shift for 1xPEG-Mal is about 3.7 kDa equivalent on average. The slower migration is likely due to the less charge on the PEG-Mal, compared to amino acids (Hara et al (2013) *Biochim Biophys Acta* 1830, 3077). To illustrate the resolution of the improved PEG-Mal labeling assay, we performed a control experiment using Cys mutant of NPR1. We expressed GFP-tagged *Arabidopsis* NPR1 (NPR1^{WT}-GFP) and its Cys-mutant (NPR1^{C82AC216A}-GFP) in *N. benthamiana* leaves. Then, the NPR1-GFP proteins were reduced by TCEP followed by labeling with PEG-Mal (in the new Fig. 5c). Non-labeled NPR1^{WT}-GFP and NPR1^{C82AC216A}-GFP showed comparable mobility, but PEGylated NPR1^{C82AC216A}-GFP exhibited an apparently smaller band than PEGylated NPR1^{WT}-GFP (in the new Fig. 5c). These data demonstrated that our PEG-Mal labeling assay has enough resolution to detect one disulfide bond formation (or reduction) in 19 Cys residues. Using this improved method, we designed a new experimental workflow (in the new Fig. 5d) to detect free Cys residues (in the new Fig. 5e) and disulfide-bond-forming Cys residues (in the new Fig. 5f). We think that our detection system is sensitive enough to detect both the free and the disulfide-bond-forming Cys residues. Importantly, and consistent with our original data, our new data also support that NPR1-GFP has no disulfide bond regardless of SA treatment.

Moreover, because PEG-Mal labelled NPR1-GFP already runs at an unexpected higher molecular weight, this is likely not a suitable assay to determine Cys oxidation of NPR1-GFP. Hence the assay is inconclusive.

> As discussed above, 19xPEG-Mal will be equivalent to 70 kDa shift in an SDS gel. NPR1-GFP runs around 90 kDa (in old Fig 5c, and new Fig. 5c, e, f) and thus we expected PEG-Mal labelled NPR1-GFP to run around 160 kDa, which we observed consistently. Similarly, PR1 was PEGylated only when the sample had been reduced with TCEP prior to PEG-Mal labeling, demonstrating that PEG-Mal selectively reacts with free thiol in cysteine residues of tested protein samples in our system (new Fig. 5e, f). Thus, we believe that our new PEG-Mal assay is conclusive enough to determine the Cys-oxidation status of NPR1.

(11) Lines 262-264: the described data that DTT diminished the 250kDa signal is not shown in Fig

6a or anywhere else.

>We are sorry that we were not clear on this in our previous text. The second panel from the top in Fig. 6a (anti-NPR1_+DTT) is the corresponding data. The 250 kDa signal was observed in non-reducing condition (top panel in Fig. 6a; anti-NPR1_-DTT) but not in reducing condition (Second panel). To make it clearer, we have now mentioned this in the corresponding text.

(12) Fig 6a The fact that TNX induces NPR1 monomer either on its own or in combination with SA creates more questions than answers. This does not square up with the effect of TNX on cellular redox.

>Thank you for your comments. Indeed, we also find it difficult to interpret such observation based on the previous oligomer-monomer transition model of NPR1 (Mou et al., 2003). Based on our data in Fig. 6a, b, which show that NPR1 oligomerizes *via* intermolecular disulfide bonds *in vitro*, but not *in vivo*, we would like to offer a possible explanation for our data, independent from Mou et al., (2003) model. As we have previously explained in the original Discussion, the reason why SA-induced NPR1-GFP monomer appearance *in vitro* could be due to less accessibility of free thiols in NPR1 *in vitro* (in the new Discussion, line 452-462). TNX has a similar effect with SA on the appearance of NPR1-GFP monomer (Fig. 6a). We postulate that this *in vitro* effects may be potentially due to high oxidized glutathione (GSSG) contents. It was previously shown that GSSG can react with free thiols *via* a thiol-disulfide exchange between GSSG and a cysteine thiolate of a protein, resulting in the formation of GSH and R-SSG, if GSSG is in abundant in solution (Bak and Weerapana (2015) Mol Biosyst 11, 678). Thus, it is plausible that NPR1-GFP reacts with GSSG *in vitro*, given that we observed a higher amount of GSSG in the presence of TNX (Fig. 5b).

(13) The data in Fig 6c are quite surprising, as a similar experiment has been reported by Mou et al. (2003) who showed that NEM and IOD did not affect NPR1-GFP oligomer detection. The effect of NEM in this figure is particularly surprising as the Methods indicate that the protein sample was not incubated and kept at 4C. NEM is not very reactive at low temperatures and requires at least incubation at room temperature or higher to block Cys residues. Nonetheless, the authors observe an oligomeric signal in the +NEM samples (albeit lower than in the -NEM samples), so it seems premature to conclude that NPR1 oligomerisation occurs only *in vitro* and not *in vivo*.

> Thank you for your comments. This is indeed a big surprise to us, as this is the holy grail of the report by Mou et al. In the original Fig. 6c experiment, if NPR1 contains reactive Cys residues, both spontaneous redox reactions of Cys residues and blocking by NEM will proceed in a competitive

manner during the protein extraction. Importantly, the spontaneous redox reactions, such as oxidation, is a fast reaction that can proceed on the microsecond timescale (Creighton (1984) *Methods Enzymol* 107, 305) and have a potential to mask the NEM effect. Therefore, we adopted lower temperature conditions that minimize the rate of redox-related modification of Cys residues (Fava et al., (1957) *J. Am. Chem. Soc.* 79, 833) and we believe that this is the critical condition for the assessment. Although the reactivity of NEM is temperature sensitive as reviewer#1 pointed, this reaction still proceeds even on ice (Russo et al (2020) *Anal Bioanal Chem* 412, 1639). Indeed, as showed in the new Fig. 6b experiment, we observed a clear decrease in the amount of NPR1 oligomer in the presence of NEM, indicating that cysteine thiols in NPR1 are effectively blocked by NEM under our experimental condition. Interpretation of the difference in the NEM effect between Mou et al. and ours is important. As described in the Discussion, we consider that a slightly basic condition (pH 7.5, buffered by 50 mM tris) may have resulted in inefficient blocking of Cys residues in study of Mou et al. (2003). We believe that NPR1 oligomerization mainly occurs *in vitro*, but as the reviewer pointed out, we cannot completely exclude the possibility of small amounts of NPR1 oligomerization *in vivo* if it's below the detection level. Thus, we have now stated "However, we cannot exclude the possibility that small amounts of NPR1 oligomerize by forming intermolecular disulfide bonds *in vivo*, which may be below the detection level in the PEG-Mal labeling assay." was incorporated into the Discussion section.

Other editorial comments:

(1) Page 2 (top): In the description of targets of SA in plants and its derivatives in mammals, primary references should be used instead of reviews. Also, SA accumulates in plants cells upon recognition of biotrophic pathogens and may not be as effective against necrotrophics. The text should be revised accordingly.

>Thank you for this comment. We revised the main text as reviewer#1 pointed.

(2) Page 3 (top): "NPR1 was also reported to serve as an intrinsic link between daily redox rhythms and the circadian clock (20). Thus, NPR1 has been proposed to be a redox sensor in plants." It is not clear the relevance of this information here.

>Thank you for this comment. We deleted the sentence because it is not indispensable.

(3) Page 3 (bottom): Introduction of key terminology for this study, such as basal immunity, hypersensitive response and effector triggered immunity, would help the non-expert reader.

>We have added the description about basic terminology on plant immunity in the Introduction.

Reviewer #2 (Remarks to the Author):

The manuscript is a result of a research concept aiming at identifying chemicals that prime the plant immune response. Here, different nonsteroidal anti-inflammatory drugs (NSAIDs) were tested. Although the compound TNX primed the cell death response, it suppressed a big sector of the NPR1-dependent response towards salicylic acid. The authors went on to characterize the mechanism how TNX might interfere with the function of SA-activated NPR1 and by doing so they got contradictory results to a longstanding working model for NPR1 action.

It is well established that SA-mediated responses correlate with the activation of the anti-oxidative system, leading to higher glutathione levels and to a higher GSH/GSSG ratio. According to the current model, this redox shift leads to the reduction of intermolecular disulfide bridges in the cytosolic multimeric NPR1 complex, allowing nuclear translocation of resulting monomers. In this manuscript, it is shown, that TNX interferes with the establishment of the „reduced state“ through an unknown mechanism. Therefore, it was expected that TNX would interfere with the reduction of the NPR1 multimer to monomers.

However, TNX enforces monomer formation and acts additively with SA in this respect (Figure 6a). When preparing the extract in the presence of 10% TCA, which interferes with the oxidation of thiol groups, the oligomer is not formed any more. This result suggests that disulfide-bridged NPR1 oligomers are not present in vivo but rather form only in vitro when keeping the protein in a buffer without DTT. Pre-treatment of plants with SA or TNX would change the redox state of interaction partners of NPR1, which would protect a fraction of NPR1 against oxidation. If this were true, it would question the established model of the redox-regulation of NPR1. Figure 6c supports this conclusion: NEM treatment of plant extracts, which protects reduced thiol groups against oxidation, prevents – like DTT treatment – oligomerisation of NPR1. The third piece of evidence comes from another alkylation experiment with TCA-precipitated proteins. Here mPEG was used which adds a 2kDa moiety to each cysteine. No influence of either SA or TNX or DTT on the amount of added mPEG was observed. All these experiments point at the situation that in vivo, NPR1 seems to be in the reduced state all the time. Oligomerisation through oxidation might be an in vitro artefact and previous researchers might have been misled. To my opinion, this is the most important lesson of the manuscript. Since it would generate a little „earth quake“ in the community, experiments should not leave any doubt.

Fig. 5: Criticism on the mPEG experiment might be raised since it is not shown whether a difference of 2 or 4 kDa, which would be indicative for the oxidation of 1 or 2 residues, respectively, can be seen with this method. It would be good to perform such an experiment side by side with a construct which contains 17 rather than 19 cysteines. Maybe this can be done with recombinant protein.

>We thank for this constructive comment and suggestion of decisive experiments. We performed the experiment accordingly (See above). Because the control samples were not able to load on an identical gel due to the limitation of sample number and differences in signal intensity, we added the same ladder marker to all gels and ran these gels at the same condition (150 V, 2 h).

Moreover, the experiment should be repeated in a buffer of pH 8. When performing the reaction at pH 7, the authors cannot guarantee that all residues can be alkylated. In the worst case, the mobility of the alkylated proteins would always be the same since the potentially oxidized cysteine cannot be alkylated even after reduction because of the wrong pH of the buffer. It is unlikely, but it has to be done.

>As reviewer#2 pointed, slightly alkaline condition (pH 8.0) is suitable for well-known alkylating agents, such as iodoacetic acid or iodoacetamide. However, we consider pH 8.0 buffer to be unsuitable for Cys labeling by maleimide for the reasons described below. Firstly, maleimide reactions are specific for thiol groups in the pH range of 6.5 to 7.5, but at higher pH values, cross-reactivity with amino groups occurs (Brewer and Riehm (1967) *Anal Biochem*, 18, 248). Secondly, at pH 8.0, the proportion of deprotonated cysteine thiols will increase, which facilitates the spontaneous redox-related reactions of thiols. Thus, we think that pH 7.0 is an ideal condition for specific labeling of Cys residues by maleimide. Indeed, the PEGylated proteins converged on a single band in our PEG-Mal labeling assay (Fig. 5c, e, f) although partial PEGylation of protein ideally should result in ladder like pattern in this assay, corroborating that maleimide effectively reacts with thiols under our experimental condition.

Moreover, it has to be taken into account, that SA treatment does not completely monomerize the oligomer. Thus, it is expected that only a small portion of the protein can be alkylated to a higher degree in extracts from SA treated plants. Maybe this is below the detection level. Since the experiment has to be repeated anyhow, I recommend to use -DTT control at the same time. It would thus integrate Fig. 6b showing that -DTT does not change the mobility of NPR1. The subsequent labelling experiments with mPEG with PR1 used as a control show that TCA precipitation does not reduce an oxidized protein in the absence of DTT.

>Thank you again for suggesting good experiments. As discussed above, we re-performed the PEG-Mal labeling assay accordingly. We confirmed that TCA precipitation does not reduce disulfide bonds in PR1 (in new Fig. 5e). Again, our new data suggested that NPR1-GFP has no disulfide bond regardless of SA treatment (in new Fig. 5e). In these new experiments, we used TCEP instead of DTT because TCEP is more potent than DTT under pH 7.0 condition (Han and Han (1994) Anal Biochem 220, 5).

Minor points concerning this figure: Can the authors comment why the band for PR1 in untreated plants only shows up in the DTT-treated extracts? It would be nice to point out that the 35S-NPR1-GFP construct is in the *npr1-1* background.

>Thank you for pointing this out. We believe that signals in untreated plants represents cross-reaction with a non-specific unknown protein. The reason is that transcript levels of *PR1* were extremely low in untreated seedling (Fig. 2d) and the signal in SA-treated plants (i.e. PR1) migrated slightly further than the signal in untreated plants (i.e. non-specific unknown protein) (see for instance Fig. 4b, 4c, 5e, 6a, 6b). We added this information after *35Sp:NPR1-GFP* in line 280.

However, these experiments were done with seedlings grown in sucrose containing medium. This might have been necessary due to the drug treatment, but the experiments addressing the effect of SA on the redox state of NPR1 should also be done with soil-grown plants.

>Thank you for this comment. As shown in Fig. 4c and 6b, native NPR1 levels in mock-treated Col-0 WT is close to the detection limit, thus it is unlikely that we will detect the clear signal of NPR1 in these plants after PEG-Mal labeling. Therefore, we tested the redox state of NPR1 in soil-grown *35Sp:NPR1-GFP* plants. Consistently, our new data suggested that NPR1-GFP has no disulfide bond regardless of SA treatment even in the soil-grown *35Sp:NPR1-GFP* plants (new Supplementary Fig. 10). However, these soil-grown plants exhibited severe stunting, development of micro lesions, and PR1 accumulation without SA treatment as reported previously (new Supplementary Fig. 10; Love et al., (2012) PLoS One 7, e47535 ;Wirthmueller et al., (2018) New Phytologist 220, 232). Thus, SA-dependent defense could be activated in the soil-grown *35Sp:NPR1-GFP* plants regardless of SA treatment. In contrast, in seedling of *35Sp:NPR1-GFP* grown in liquid media, PR1 accumulation clearly shows SA-dependency, and NPR1-GFP was detectable even in mock-treated plants (new Fig. 5e). We think that *35Sp:NPR1-GFP* seedlings can offset the shortcomings of the other samples and provide convincing evidence for the redox state of cysteines in NPR1 *in vivo*.

If these experiments continue to point to the direction that NPR1 does not form disulfide bridged oligomers, this very important message should be mentioned in the abstract or even in the title. Many courses at universities all over the world teach the students how NPR1 works and part of it seems to be based on the misinterpretation of experiments.

>Thank you for supporting our work. We have now revised the abstract and emphasize this fact accordingly. We did not change the title due to the word limitation (15 words or less).

There is another discrepancy to the current model as proposed by the Dong group. In the absence of SA, NPR4 degrades NPR1. In the presence of SA, the interaction of NPR1 and NPR4 is disrupted, so that NPR1 can accumulate. If there is even more SA, NPR3 binds SA and initiates degradation of NPR1, which would otherwise interfere with cell death. If this model were true, the *npr4* mutant and the *npr3 npr4* double mutant should show elevated NPR1 levels in the absence of SA, which was the case in Fu et al Nature 2012. This was not reproduced in Figure 4 of this manuscript and this should be pointed out.

>Thank you for this comment. Now we have pointed this out in the text.

Since TNX interferes with NPR1 accumulation. One hypothesis was that TNX would maintain the interaction between NPR4 and NPR1 even in the presence of SA. This was excluded.

>Thank you for your insight. We believe this hypothesis goes against our finding in Fig. 4c and Supplementary Fig. 9. In Fig. 4c, we detected NPR1 levels in *npr3npr4* mutant in the presence of SA and/or TNX. We still observed comparable reduction in NPR1 levels in both WT Col-0 and *npr3npr4* mutant with TNX. In Supplementary Fig. 9, we performed yeast two hybrid assay between NPR1 and NPR4 in the presence of SA and/or TNX. We observed that NPR1 and NPR4 remain dissociate in the presence of SA and TNX. Both of these data indicate that TNX is unlikely to have an effect on NPR4-NPR1 interaction *per se*.

Other points.

Concerning the TNX part of the story, I have the following suggestions. Since the effect of TNX on NPR1-mediated immune responses is in the focus of the paper, Figure 1a, which addresses the priming effect of different drugs on PTO-mediated cell death, should go to the supplement.

>Thank you for this comment. We moved the data to the supplement (now Supplementary Fig. 2).

When explaining Figure 3, it should be pointed out that TNX has negative effects on NPR1-dependent gene expression even in the absence of NPR1. This seems to allow the conclusion, that other processes that happen in the absence of NPR1 are affected.

>Thank you for this comment. We re-analyzed our transcriptome data and added the statement “However, 207 genes (32.8 %) of SA-inducible NPR1-dependent genes were suppressed upon co-treatment with TNX even in the *npr1* mutant (\log_2FC (SA/SA+TNX) in *npr1* > 1), implying that TNX also affects NPR1-independent processes (Supplementary Data 8)” into the Result.

Discussion: Line 423: I had the impression that Tada et al. claim that SAR is compromised in *trx* mutants, also PR1 expression is compromised. Still, it is feasible that TRXs are required for other processes and not for the reduction of NPR1.

>We agree with reviewer#2's opinion. We have added the statement “This work also claims that SAR and SA-induced *PR1* expression are compromised in *trx* mutants due to attenuation of NPR1 monomerization.” into the Discussion.

REVIEWERS' COMMENTS

Reviewer #2 (Remarks to the Author):

This manuscript contains essentially two parts:

One part describes the effect of the nonsteroidal anti-inflammatory drugs (NSAIDs), specifically TNX, on plant immunity using *Arabidopsis thaliana* as a model system (seedlings and adult plants). In essence, TNX primes the hypersensitive cell death and suppresses SA-dependent immune responses leading to higher susceptibility against the virulent *Pseudomonas* strain Pto. The authors describe that TNX leads to more oxidizing conditions in the cytosol which correlates well with the induction of genes related to oxidative/xenobiotic stress. At the same time, SA-mediated responses are suppressed. However, as already pointed out by reviewer #1, the authors failed to identify the molecular target of TNX. As a matter of fact, the claim that TNX can be used as a tool to understand novel aspects of plant immunity is not yet justified.

The other part of the manuscript describes a series of very valuable experiments that question the long-standing model that the SA receptor NPR1 is redox-regulated. As already pointed out in my previous review, this piece of information is important and the rebuttal letter clearly documents that the authors are very aware of the underlying chemical principles of the state of the art methods to analyze the in vivo redox state of a protein. The important take home message is that NPR1 does not form oxidized oligomers in vivo, but that the established model is based on in vitro oxidation processes in extracts prepared without any reducing compound. This artefact is less pronounced in SA-treated plants. The authors found a quite plausible explanation for this, e.g. that NPR1 is posttranslationally modified in SA-treated plants so that the in vitro oxidation events are less pronounced.

My specific criticism concerning the first part is that TNX affects most NPR1-dependent promoters also in the absence of NPR1. Thus, it seemed likely from the beginning on that an NPR1-independent mechanism is affected directly or indirectly by TNX. This could be for instance a subunit of a mediator complex that controls these promoters, but indeed, also maybe TGA transcription factors which might yield basal expression in the absence of NPR1. I honestly do not agree with the conclusion that TNX affects NPR1 activity as stated in lanes 403 and 405

„Thus, TNX likely interferes with NPR1 activation, in addition to modulating NPR1 levels in response to cellular redox shifts“

It is not the author's fault that there still many conflicting results and unexplained observations on NPR1 action. These become evident in this manuscript as well. For instance: NPR1 protein levels increase if NPR1 is expressed under its own promoter, although transcript levels do not increase to the same extent. However, in lines that constitutively express this increase in protein levels is not observed. This has been explained by high NPR1 levels in the 35S plants, but in this article, protein levels are not different in the different lines (Fig. 6b). However, in view of these unexplainable results, it is also unexplained how TNX interferes with NPR1 protein accumulation when NPR1 is expressed under its own promoter but not when it is expressed under the 35S promoter.

I also do not agree with the statement in the abstract (line 34)

„Therefore, oxicam-type NSAIDs highlight the importance of SA on cytosolic redox status....“

Criticism: So far, there is a correlation between SA-induced reduction of the cytosol and gene expression. If this reduction is inhibited by TNX, SA-mediated gene expression does not occur. This is just a correlation. It could well be that SA-mediated reduction of the cytosol is not required for SA signaling, since binding of SA to NPR1 might be the crucial event and TNX might affect other processes affecting a subgroup of NPR1-dependent promoters (see above).

The second part is of uppermost relevance and should be published in a high ranking journal, but I do understand that the authors would like to hide this statement due to possible consequences on reviewing processes affecting their other publications. It might well be that experts in this field of research already have doubts on the model of redox-regulated NPR1, but nobody takes the risk to publish results against highly cited papers that even led to follow-up articles in other prestigious journals.

Therefore, I suggest that the authors re-consider their conclusions on the TNX-part and rephrase the abstract accordingly.

Reviewer #3 (Remarks to the Author):

This manuscript by Ishihama et al. identified the oxicam-type NSAIDs tenoxicam (TNX), meloxicam, and piroxicam as inhibitors of immunity to bacteria and SA-dependent plant immune response. Further analysis showed that TNX treatment induces oxidation of cytosolic redox status and decreases NPR1 levels. Furthermore, NPR1 was shown to be present predominantly in the reduced form in vivo regardless of SA or TNX treatment. This is surprising as previously it was reported that under normal growth condition NPR1 exists predominantly in cytoplasm as oligomers formed via redox-sensitive intermolecular disulfide bonds between cysteine residues. Although the manuscript is a bit short on elucidating the exact mechanism of how TNX affects NPR1 protein levels, it clarifies on whether reduction of NPR1 plays a role in SA perception and signaling, which helps to clear up some long-standing misinterpretation in the field. I think the authors have adequately addressed the concerns raised by reviewer #1 and the manuscript is suitable for publication in Nature Communication. I only have two small questions related to the mode of action of TNX on avrRpm1-induced cell death and NPR1 protein levels, which the authors might be able to address in the Discussion section, or with experimental data they might already have.

1. Can TNX enhance avrRpm1-induced cell death in npr1 mutant plants?
2. Can TNX directly bind to NPR1 in vitro?

Dear editor and reviewers,

We would like to thank the reviewers for their insightful comments, which have helped us to improve our manuscript. We have provided a point-by-point reply to address specific concerns that each reviewer had, and addressed the points raised.

We sincerely hope that you will find our revised manuscript acceptable for publication.

On behalf of all the authors,

Nobuaki Ishihama and Ken Shirasu

REVIEWER COMMENTS

Reviewer #2 (Remarks to the Author):

This manuscript contains essentially two parts:

One part describes the effect of the nonsteroidal anti-inflammatory drugs (NSAIDs), specifically TNX, on plant immunity using *Arabidopsis thaliana* as a model system (seedlings and adult plants). In essence, TNX primes the hypersensitive cell death and suppresses SA-dependent immune responses leading to higher susceptibility against the virulent *Pseudomonas* strain Pto. The authors describe that TNX leads to more oxidizing conditions in the cytosol which correlates well with the induction of genes related to oxidative/xenobiotic stress. At the same time, SA-mediated responses are suppressed. However, as already pointed out by reviewer #1, the authors failed to identify the molecular target of TNX. As a matter of fact, the claim that TNX can be used as a tool to understand novel aspects of plant immunity is not yet justified.

The other part of the manuscript describes a series of very valuable experiments that question the long-standing model that the SA receptor NPR1 is redox-regulated. As already pointed out in my previous review, this piece of information is important and the rebuttal letter clearly documents that the authors are very aware of the underlying chemical principles of the state of the art methods to analyze the *in vivo* redox state of a protein. The important take home message is that NPR1 does not form oxidized oligomers *in vivo*, but that the established model is based on *in vitro* oxidation processes in extracts prepared without any reducing compound. This artefact is less pronounced in SA-treated plants. The authors found a quite plausible explanation for this, e.g. that NPR1 is posttranslationally modified in SA-treated plants so that the *in vitro* oxidation events are less pronounced.

>We thank the reviewer for acknowledging the importance of this study.

My specific criticism concerning the first part is that TNX affects most NPR1-dependent promoters also in the absence of NPR1. Thus, it seemed likely from the beginning on that an NPR1-independent mechanism is affected directly or indirectly by TNX. This could be for instance a subunit of a mediator complex that controls these promoters, but indeed, also maybe TGA transcription factors which might yield basal expression in the absence of NPR1. I honestly do not agree with the conclusion that TNX affects NPR1 activity as stated in lines 403 and 405

„Thus, TNX likely interferes with NPR1 activation, in addition to modulating NPR1 levels in response to cellular redox shifts“

>Thank you for your insightful comments. As requested, we eliminated the statement “Thus, TNX interferes with NPR1 activation, in addition to modulating NPR1 levels in response to cellular redox shifts” from the Discussion section.

We also thank you for pointing out the possibility that TNX could target transcriptional machineries regulated by SA and thus could indirectly affect NPR1-mediated signaling pathway. To account for this idea, we have edited the statement, from “any potential molecular target(s) are regulators of cellular redox homeostasis or antioxidant enzyme(s).” to “TNX directly or indirectly interferes with regulators of cellular redox homeostasis or antioxidant enzyme(s).” in the Discussion section.

It is not the author’s fault that there still many conflicting results and unexplained observations on NPR1 action. These become evident in this manuscript as well. For instance: NPR1 protein levels increase if NPR1 is expressed under its own promoter, although transcript levels do not increase to the same extent. However, in lines that constitutively express this increase in protein levels is not observed. This has been explained by high NPR1 levels in the 35S plants, but in this article, protein levels are not different in the different lines (Fig. 6b). However, in view of these unexplainable results, it is also unexplained how TNX interferes with NPR1 protein accumulation when NPR1 is expressed under its own promoter but not when it is expressed under the 35S promoter.

> Thank you for your understanding with regard to the difficulties of explaining NPR1 protein level due to current incomplete understanding of NPR1 protein dynamics.

I also do not agree with the statement in the abstract (line 34)

„Therefore, oxicam-type NSAIDs highlight the importance of SA on cytosolic redox status....“

Criticism: So far, there is a correlation between SA-induced reduction of the cytosol and gene

expression. If this reduction is inhibited by TNX, SA-mediated gene expression does not occur. This is just a correlation. It could well be that SA-mediated reduction of the cytosol is not required for SA signaling, since binding of SA to NPR1 might be the crucial event and TNX might affect other processes affecting a subgroup of NPR1-dependent promoters (see above).

> Thank you for your comments. We have removed the description pointed out by the reviewer from the Abstract section. The last sentence of the introduction, which is likely to be the subject of this criticism, has also been removed. In addition, we added the statement “It is interesting that both SA and TNX alter cellular redox balance, but the precise relevance between oxidation of cellular redox state by TNX and its inhibitory effect on SA signaling remains to be determined.” to the Discussion section.

The second part is of uppermost relevance and should be published in a high ranking journal, but I do understand that the authors would like to hide this statement due to possible consequences on reviewing processes affecting their other publications. It might well be that experts in this field of research already have doubts on the model of redox-regulated NPR1, but nobody takes the risk to publish results against highly cited papers that even led to follow-up articles in other prestigious journals.

>We appreciate the reviewer’s kind words and encouraging comments.

Therefore, I suggest that the authors re-consider their conclusions on the TNX-part and rephrase the abstract accordingly.

>We have edited our conclusions on the TNX-part and rephrased the abstract as requested.

Reviewer #3 (Remarks to the Author):

This manuscript by Ishihama et al. identified the oxicam-type NSAIDs tenoxicam (TNX), meloxicam, and piroxicam as inhibitors of immunity to bacteria and SA-dependent plant immune response. Further analysis showed that TNX treatment induces oxidation of cytosolic redox status and decreases NPR1 levels. Furthermore, NPR1 was shown to be present predominantly in the reduced form in vivo regardless of SA or TNX treatment. This is surprising as previously it was reported that under normal growth condition NPR1 exists predominantly in cytoplasm as oligomers formed via redox-sensitive intermolecular disulfide bonds between cysteine residues. Although the manuscript is a bit short on elucidating the exact mechanism of

how TNX affects NPR1 protein levels, it clarifies on whether reduction of NPR1 plays a role in SA perception and signaling, which helps to clear up some long-standing misinterpretation in the field. I think the authors have adequately addressed the concerns raised by reviewer #1 and the manuscript is suitable for publication in Nature Communication.

>We thank the reviewer for acknowledging the importance of this study.

I only have two small questions related to the mode of action of TNX on *avrRpm1*-induced cell death and NPR1 protein levels, which the authors might be able to address in the Discussion section, or with experimental data they might already have.

1. Can TNX enhance *avrRpm1*-induced cell death in *npr1* mutant plants?

>Thank you for this comment. We have not tested the TNX effect on *Pto avrRpm1*-induced cell death in *npr1* plants. Thus, we added the statement “Therefore, we would expect that potentiation of the *Pto avrRpm1*-induced cell death by TNX should be lower in the *npr1* mutant compared with WT, although this remains to be tested.” to the Discussion section.

2. Can TNX directly bind to NPR1 in vitro?

>Thank you for this comment. We have not tested the molecular interaction between NPR1 and TNX, thus we don't know whether NPR1 can directly bind to TNX or not. Because TNX affected a subset of genes even in *npr1* mutants in our transcriptome analysis (Fig. 3), we expect that NPR1 is not the direct target of TNX. Therefore, we added the statement “Our transcriptome data revealed that TNX suppresses a subset of SA-inducible NPR1-dependent genes even in *npr1* mutants. This indicates that a potential target of TNX may regulate these genes in an NPR1-independent manner and likely rules out NPR1 as the direct target of TNX.” to the Discussion section.